



# Volumetric evolution of supraglacial lakes in southwestern Greenland using ICESat-2 and Sentinel-2

Tiantian Feng[1,2], Xinyu Ma[1,2], Xiaomin Liu[3]

[1]College of Surveying and Geo-Informatics, Tongji University, Shanghai 200092, China;

[2]The Center for Spatial Information Science and Sustainable Development Applications, Tongji University, Shanghai 200092, China;

[3]Zhejiang Agriculture and Forestry University, 666 Wusu Street, Hangzhou, 311300

*Correspondence to*: Tiantian Feng (fengtiantian@tongji.edu.cn)

**Abstract.** Surface meltwater runoff has been the primary factor affecting the trends and interannual variations in the mass

balance of the Greenland Ice Sheet. During the melting season, large amounts of surface meltwater accumulate in low-lying areas, forming supraglacial lakes (SGLs). Quantitatively characterizing the spatial and temporal changes in the volume of SGLs can provide further insights into the surface mass balance changes of the ice sheet during the melt season. In this paper, we propose a method for estimating the volume of SGLs by combining optical imagery (Sentinel-2) and satellite altimetry data (ICESat-2). First, the area of SGLs is extracted using a Random Forest (RF) model based on spectral features from Sentinel-2

imagery, achieving an Intersection over Union (IoU) of 90.20% compared to manually delineated lake extents. Second, the depth of SGLs along the ICESat-2 profile is detected using the kernel density analysis method. Finally, a multi-layer perceptron (MLP) model constructs the nonlinear relationship between the reflectance ratio from Sentinel-2 imagery and the depth of SGLs detected by ICESat-2 data. The accuracy of depth inversion based on the MLP model surpasses traditional empirical formula methods, achieving a mean absolute error of 0.42 m. The trained MLP model is then used to estimate the depth over

the entire lake areas. The proposed volume estimation method for SGLs is applied to southwestern Greenland, capturing the volumetric evolution of SGLs throughout the entire melt season of 2022. The results reveal significant variations in the distribution, area, depth, and volume of SGLs throughout the melt season. Initially, SGLs form along the coastlines and later spread inland, expanding in area and depth. The maximum total volume of SGLs is reached on August 1st, amounting to 9.30 $\times 10^8$ m³. Afterward, SGLs above 1200 m continue to increase in volume, while SGLs below 1200 m begin to decrease. In late

August, as the melt season draws to a close, SGLs diminish and retreat to coastal regions, with a notable reduction in volume. Additionally, according to the evolution characteristics of SGLs at different elevations, SGLs above 800 m exhibit a similar evolution pattern. A temporal discrepancy is observed in the attainment of maximum values for both mean area and mean depth, implying a differential rate of development of SGLs in the horizontal and vertical dimensions. The elevation range of 1200 m to 1600 m is the most favorable for the evolution of SGLs.



## 1 Introduction

The Greenland Ice Sheet, the second largest ice sheet in the world, exerts a significant influence on global sea levels (Shepherd et al., 2020). Since the 1990s, remote sensing observations have revealed a pronounced acceleration in the melting of the Greenland Ice Sheet (Mouginot et al., 2019; Slater et al., 2020). Research indicates that surface meltwater runoff has been the primary factor affecting the trends and interannual variations in the mass balance of the Greenland Ice Sheet between 2000 and 2019 (Box et al., 2022). Throughout the melt season, a substantial volume of surface meltwater accumulates in depressions, forming supraglacial lakes (SGLs). These SGLs, integral to the surface hydrological system of the Greenland Ice Sheet, eventually discharge into the ocean or infiltrate beneath the ice through various pathways, such as surface runoff, crevasses, or moulins (Meierbachtol et al., 2013; Poinar and Andrews, 2021; Smith et al., 2015), thus significantly influencing the mass-energy balance of the ice sheet (Arthur et al., 2020; Pope et al., 2016). Quantitatively characterizing the spatial and temporal changes in the volume of SGLs can provide further insights into the surface mass balance changes of the ice sheet during the melt season (Banwell et al., 2012).

The area and depth of SGLs are crucial parameters for estimating their volume. The Normalized Difference Water Index for ice ($NDWI_{ice}$) proposed by Yang and Smith (2013), calculated using red and blue bands, is an effective index for identifying water features in ice and snow conditions. By applying a predetermined threshold, supraglacial water bodies can be effectively highlighted. Moreover, a variety of machine learning and deep learning methods, such as Random Forest (RF), Support Vector Machine (SVM), U-net, and Convolutional Neural Networks (CNN), have also been employed in the area extraction of SGLs (Chouksey et al., 2021; Hu et al., 2022; Jiang et al., 2022; Lutz et al., 2023; Yuan et al., 2020). These methods have demonstrated high accuracy and produced favorable outcomes in the area extraction of SGLs.

In contrast, compared to the area extraction of SGLs, the observation of lake depth faces greater challenges, which is also a crucial reason leading to inaccurate volume estimation, unreliable seasonal meltwater accumulation estimation, and difficulty in analyzing the formation and drainage events (Melling et al., 2024). To date, the primary methods for obtaining the depth of SGLs include field measurements and remote sensing inversion. Although field depth measurement is the most accurate method, the harsh environment in Greenland makes such measurements labor-intensive and limited in scope, resulting in sparse data coverage insufficient for large-scale scientific research. The primary remote sensing data sources for extracting the depth of SGLs in Greenland include optical remote sensing imagery and satellite altimetry data. Optical remote sensing imagery can comprehensively cover entire lakes and has high revisit rates, offering a relatively complete time series for observing lake changes. To estimate the depth of SGLs using optical remote sensing imagery, the radiative transfer equation proposed by Philpot (1989) is usually adopted (Pope et al., 2016; Williamson et al., 2018), which establishes a relationship between water depth and band reflectance based on the physical properties of light attenuation. On the other hand, there are also methods for lake depth retrieval through parameter fitting, which combine in-situ measurement data with optical imagery single band reflectance to fit parameters in an empirical formula, establishing a nonlinear relationship between lake depth and band reflectance (Box and Ski, 2007). Moreover, Legleiter et al. (2009, 2014) proposed the optimal band ratio analysis (OBRA)



algorithm, identifying a linear relationship between the logarithmic value of the ratio of reflectance values in the green and red bands and lake depth. Although these methods can achieve large-scale lake depth extraction, it is necessary to parameterize all

of the aforementioned variables. Furthermore, assuming a predetermined relationship between lake depth and spectral observations carries significant uncertainty when applied across different regions, sensors, and attenuation rates caused by variations in lake water composition (Melling et al., 2024).

The launch of the Ice, Cloud, and Land Elevation Satellite-2 (ICESat-2) in 2018 has provided a new data source for the inversion of SGL depths. The laser beams of ICESat-2 have penetrative capability, enabling the acquisition of photons reflected

from both the surface and bottom of SGLs along the laser beam's path(Jasinski et al., 2021). By measuring the height difference between the surface and bottom photons, the depth of SGLs can be calculated. Fair et al. (2020) proposed the Lake Surface–Bed Separation (LSBS) algorithm based on ICESat-2 data, which separates the surface and bottom photons of SGLs by a predefined depth range, successfully extracting the depth of SGLs. However, the LSBS algorithm is not automated due to differences in predefined depth between lakes. Based on the multi-layer photon reflection characteristics in ICESat-2 data,

fully automated algorithms, such as the Watta algorithm (Datta and Wouters, 2021) and the automated location and depth retrieval (ALD) algorithm (Xiao et al., 2023), use kernel density estimation methods to estimate the surface and the bottom, thus avoiding parameter selection for each lake. These data-driven depth inversion methods based on ICESat-2 data can provide high-precision elevation data for estimating the depth of SGLs (Fricker et al., 2021; Lutz et al., 2024; Melling et al., 2024). Nevertheless, the limited distribution of the ICESat-2 tracks results in limited coverage of the inversed depth of SGLs.

Recently, there has been research on the inversion of SGL depths by combining altimetry data and optical imagery to compensate for the limitations of individual data sources. Ma et al. (2020) refined ICESat-2 data by using the Density-Based Spatial Clustering of Applications with Noise (DBSCAN) algorithm, then trained the empirical model by fitting parameters in linear band models and band ratio models separately, and applied the resulting models to Sentinel-2 imagery for shallow water depth retrieval. Thomas et al. (2021) utilized ICESat-2 bathymetric photons and Sentinel-2 imagery to generate bathymetric

maps for nearshore coastal areas. Machine learning methods have also been increasingly applied in water depth inversion. Lai et al. (2022) proposed a multilayer perceptron (MLP) model that uses lake depths extracted by ICESat-2 data as a reference, combined with the spectral information from Landsat-8, to invert the depth of several shallow water regions in mid-to-low latitudes. For high-latitude SGL depth extraction, the amount of available data is significantly reduced compared to low- and mid-latitudes, making direct application of the low- and mid-latitude depth extraction model less effective (Lv et al., 2024).

Lv et al. (2024) combined ICESat-2 and Sentinel-2 data, utilizing a backpropagation (BP) neural network to extract the depth of SGLs in southwestern Greenland. They conducted depth extraction and method validation in a small area and compared the changes in SGLs during the same period from 2019 to 2023. The use of machine learning combined optical images and altimetry data to obtain the depth of SGLs has been tentatively attempted on a small scale at high latitudes, and the application on a large scale needs to be further analyzed. Moreover, existing studies predominantly examine the interannual variations of

SGLs, while the intra-seasonal changes during a single melt season have not been adequately addressed.




In this paper, we propose a method for inverting the depths of SGLs on the Greenland Ice Sheet by combining optical imagery (Sentinel-2) and satellite altimetry data (ICESat-2), intending to combine the accuracy of altimetry data with the comprehensive coverage of optical images. First, the area of SGLs is extracted from the Sentinel-2 imagery using the RF algorithm, allowing for rapid localization of lake areas along ICESat-2 tracks. Within these areas, lake depths along ICESat-2

tracks are detected based on kernel density analysis algorithm. Subsequently, a MLP is employed to establish relationships between lake depths and various spectral features of SGLs, specifically the ratio between different bands' spectral reflectance. This allows for the inversion of lake depths outside of the ICESat-2 tracks. The proposed method is applied in the southwestern region of Greenland, capturing the spatiotemporal changes in lake areas, depths, and volumes over multiple periods within the 2022 melt season. By analyzing the variations in area, depth, and volume of SGLs, the substantial fluctuations that occur within

a single melt season is highlighted. Furthermore, the characteristics of SGLs at different elevations is compared, which offering valuable support for ongoing research on surface melting processes in Greenland. This detailed examination of SGL dynamics contributes to a better understanding of the impacts of climate change on polar regions, emphasizing the necessity for continuous monitoring and analysis in these sensitive and vast areas.

## 2 Study Area and Data

### 2.1 Study Area

The study area is located on the southwest coast of Greenland as shown in Fig.1, which has shown a significant melting trend over the past few decades (Van Den Broeke et al., 2016). It is bounded by the southwest drainage subsystem No. 6.2 (Zwally et al., 2012), covering a total area of 136,902 km². During the melt season, typically between June and August, this region has an active supraglacial hydrological system (Hu et al., 2022). The excess runoff originates from low-lying (< 2000 m a.s.l.)

parts of the ice sheet (Gledhill and Williamson, 2018), where SGLs normally occur. The total area of SGLs in the southwest region is the largest among all other regions (Hu et al., 2022), and the formation and drainage of SGLs in this region have a great impact on the surface mass balance of the southwest region of Greenland (Zhang et al., 2023).



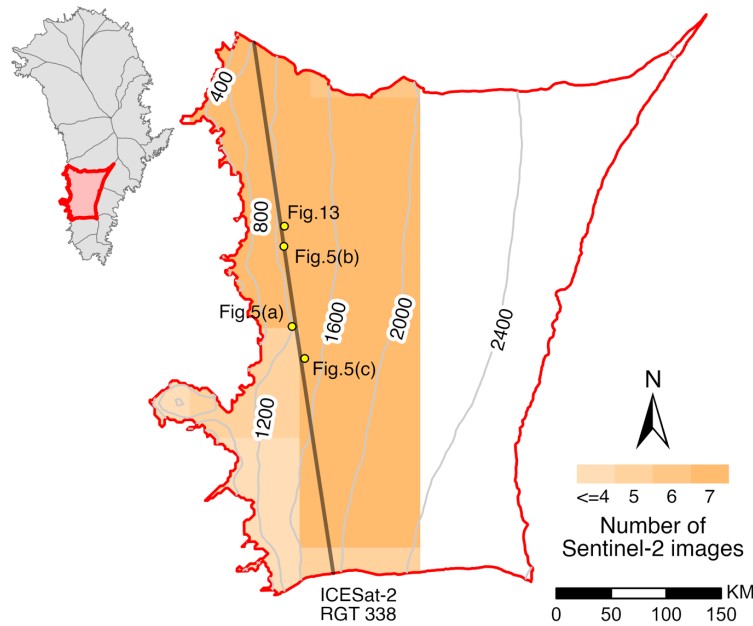

**Figure 1. Study area. Contour lines calculated from ArcticDEM are visible as grey lines. Yellow points indicate the locations of the lakes in the study area, as shown in Fig. 5 and Fig. 13.**

## 2.2 Sentinel-2 Imagery

Sentinel-2 consists of two polar-orbiting satellites (Sentinel-2A and Sentinel-2B), and the dual-satellite operation allows Sentinel-2 image data to provide a revisit cycle of about 5 days. The MultiSpectral Instrument (MSI), a push-broom optical sensor on board Sentinel-2, can collect data in 13 spectral bands. Using the significant attenuation characteristics of light in water, four bands with a spatial resolution of 10 m are used in this study, specifically the three visible light bands (Blue band with a center wavelength of 0.49 μm; Green band with a center wavelength of 0.56 μm; Red band with a center wavelength of 0.665 μm) and one near-infrared (NIR) band with a center wavelength of 0.842 μm. Considering that SGLs occur only below the equilibrium line altitude, only images covering areas below 2000 m within the basin are selected. To minimize the effect of cloud cover, we limit the cloud cover when selecting the images. During the whole melt season, when the SGLs of Greenland undergo the most pronounced changes, a total of 80 Sentinel-2 level-1C images (ortho- and geometrically corrected Top-of-Atmosphere (TOA) reflectance products) divided into 7 periods, with a period of approximately ten days between observations, are included in this study, covering the study area from June 7 to August 28, 2022. It should be noted that the single-day image, sometimes, may not cover the entire study area, such as the period between June 15 and June 20, and the period between June 30 and July 4. Therefore, we use multiple-day images to compose these two periods, denoted by June 17 and July 2 to represent these two periods in the following text.





### 2.3 ICESat-2

ICESat-2 was launched by the National Aeronautics and Space Administration (NASA) in September 2018 with polar exploration as its primary objective. It can provide elevations of sea ice, land ice, forest canopies, water height, urban areas,

etc.(Neumann et al., 2021). Equipped with the topographic laser altimetry system Advanced Topographic Laser Altimeter System (ATLAS), it is capable of transmitting laser pulses with a wavelength of 532 nm at a repetition frequency of 10 kHz. ATLAS employs three pairs of laser pulses, with each pair separated by approximately 3 km in the cross-track direction. The satellite acquires overlapping light spots with an interval of approximately 0.7 m and a diameter of approximately 17 m along its orbit with a 91-day revisiting cycle.

In this study, strong beams in each pair of laser pulses are selected. The ATL03 product provides photon data of surface elevation with latitude and longitude coordinates using the WGS84 ellipsoid as the reference ellipsoid with a spatial resolution of 0.1 m and 0.7 m in horizontal and vertical directions, respectively, which is mainly used to extract the depth of the SGLs. The ATL06 product provides accurate land ice surface elevation information with geolocation at a spatial resolution of 20 m, which is used to exclude significant height noise. Both ATL03 and ATL06 products are downloaded through the National

Snow and Ice Data Center website (https://nsidc.org/data/icesat-2). Considering the changes in SGLs, it is important to minimize the temporal difference between ICESat-2 data and Sentinel-2 images. Therefore, the ATL03 product, and the corresponding ATL06 product acquired from the orbit of the Reference Ground Track (RGT) No. 338 on July 14, 2022, are utilized in this study, since it's the only day during the entire melt season of 2022 when both ICESat-2 data and Sentinel-2 images are available. Then, both ICESat-2 data and Sentinel-2 images are converted to the UTM zone 22N (EPSG:32622) for

further analysis.

### 3 Methods

The proposed framework for the inversion of SGLs, as shown in Fig. 2, consists of three modules: extraction of SGLs using Sentinel-2 imagery, detection of SGLs' depths on the ICESat-2 RGT, and inversion of the entire lake depths using a MLP

model.



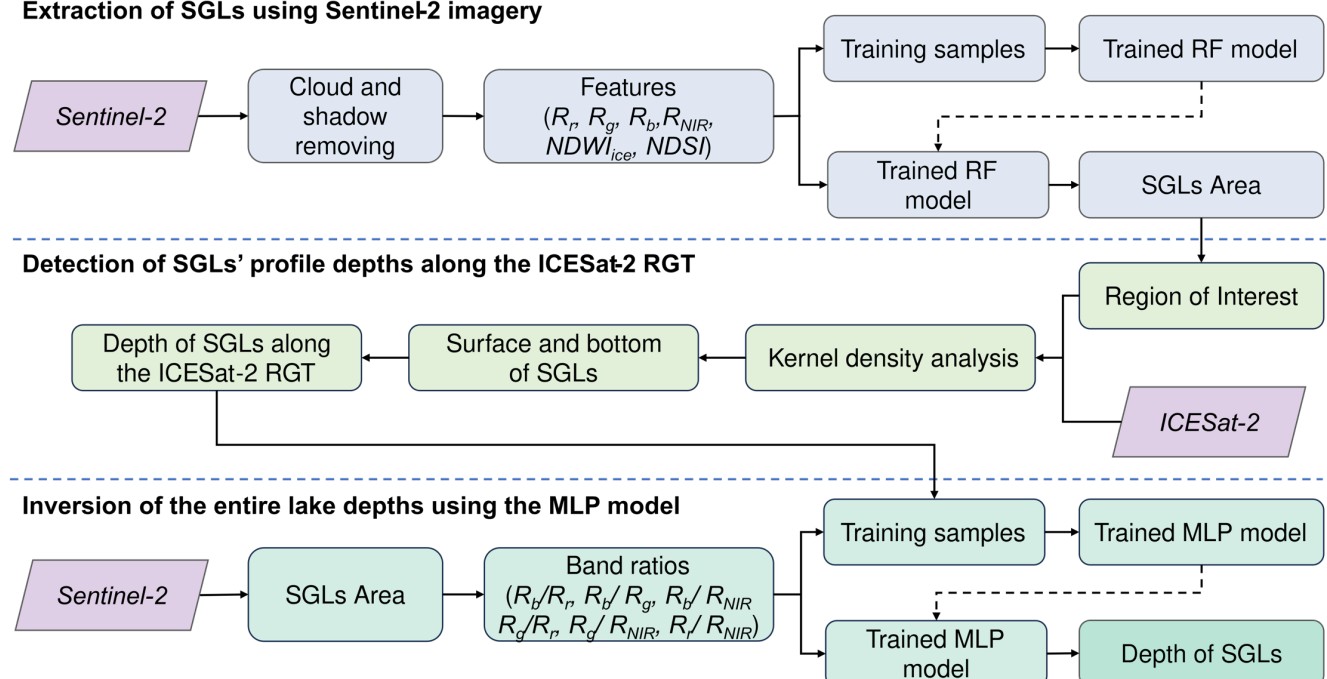

**Figure 2. Framework of proposed SGLs' depth inversion method.**

**3.1 Extraction of SGLs using Sentinel-2 imagery**

Firstly, the cloud and shadow pixels on the Sentinel-2 image are removed based on the quality tag of the QA60 band to
eliminate their impact on the extraction of SGLs. Then, the RF model, an ensemble learning method composed of multiple independently trained decision trees, is utilized to extract the lakes based on the spectral features. The advantage of the RF model is its ability to reduce the risk of overfitting by constructing decision trees using randomly selected samples and features (Breiman, 2001). The final prediction result of an RF is based on the majority vote from all decision trees since each decision tree is independent and complements each other. For feature selection, in addition to the reflection values of the red, green,
blue, and NIR bands, $NDWI_{ice}$ and the Normalized Difference Snow Index ($NDSI$) (Hall et al., 1995) are also included, considering the unique icy and snowy environment of SGLs. The calculation for $NDWI_{ice}$ and $NDSI$ is shown in equations (1) and (2), where $Rr$, $Rg$, $Rb$, and $R_{NIR}$ represent the reflection values of the red, green, blue, and near-infrared bands, respectively.

$$NDWI_{ice} = \frac{R_b - R_r}{R_b + R_r} \tag{1}$$

$$NDSI = \frac{R_g - R_{NIR}}{R_g + R_{NIR}} \tag{2}$$

The RF model is trained using the training samples, and the trained model is then applied to Sentinel-2 images of seven periods in the study area to extract SGLs. All these Sentinel-2 image processing tasks are conducted on the Google Earth Engine (GEE) platform.



## 3.2 Detection of SGLs' profile depths along the ICESat-2 RGT

The regions of interest in the ICESat-2 data are identified based on the extraction results of SGLs from the Sentinel-2 image.
To ensure that the altimetry data further processed includes both data inside and outside the lake, a 100-meter buffer zone around each SGL is established. Windows based on ATL06 surface elevation data establish the vertical extent of the data used, while buffer zones determine the range of data along the track direction. Then, the kernel density analysis method is employed to discern the surface and bottom of the lake from the multiple reflections of photons on the various surfaces of the SGL. Subsequently, the precise boundary of the SGL is determined according to the breakpoints of the surface slope change.
Considering that the surface of the ice sheet has a more pronounced topographic undulation, while the SGL is water surface and therefore horizontal, the extent of the SGL is determined by detecting the slope change of the surface. Within this precise boundary of the SGL, a refraction correction (Parrish et al., 2019) is applied to determine the actual depth of SGL. The most reliable method to assess uncertainty of depth oriented from ICESat-2 data is by comparing inversion results with in-situ measurement. However, due to the harsh environment in Greenland and the rapid changes of SGLs during the melting season,
obtaining in-situ data near the ICESat-2 transit time is challenging. In this study, the quality of depth data derived from ICESat-2 is ensured through visual inspection. And the profile depth extracted from ICESat-2 ATL03 data is considered as the reference for the Sentinel-2 depth estimation.

## 3.3 Inversion of the entire lake depths using the MLP model

Considering the discrepancy in spatial resolution between the depth estimates obtained by ICESat-2 data and the Sentinel-2
imagery, we create a dataset by computing the average depth value from the output of ICESat-2 ATL03 data within a Sentinel-2 pixel. Inspired by Lai et al. (2022) on optical shallow water depth inversion in mid and low-latitude regions, we construct a MLP architecture consisting of three hidden layers (Fig. 3): the first with 128 nodes, the second with 32 nodes, and the third with 16 nodes. The three hidden layers are connected using the rectified linear unit (ReLU) activation function. A linear activation function is applied to the output layer, which provides depth estimates corresponding to each pixel. The depth of
SGL exhibits a non-linear relationship with the ratio of reflectance between blue and red bands (Legleiter et al., 2014). Motivated by this insight, we opted for a more comprehensive approach by considering multiple band ratios, specifically the top-of-atmosphere (TOA) reflectance ratios between the red, green, blue, and near-infrared bands, derived from Sentinel-2 imagery, as input features for the MLP. Both the depth of SGLs from two pairs of ICESat-2 RGT No. 338 and the corresponding band ratios within SGL areas from Sentinel-2 images are used to train the MLP model, and the data in the
remaining pair of ICESat-2 RGT No. 338 are used to test the performance of the MLP model. Then, the trained MLP model is applied to invert the depth of SGLs within seven periods in the whole study area.



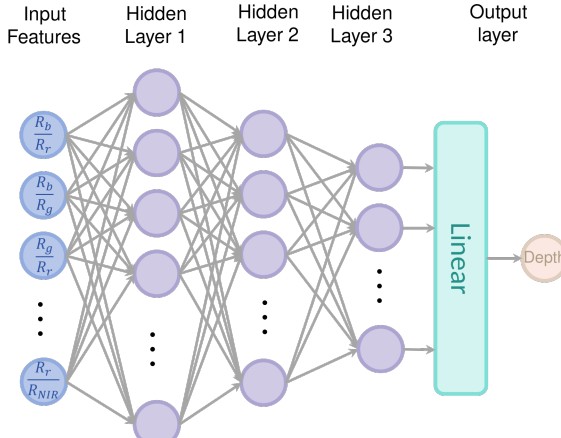

**Figure 3. Structure of MLP model.**

# 4 Results

## 4.1 Evaluation of area extraction and depth inversion

For each time period, we randomly sampled 50 pixels from both SGL areas and other areas in the mosaiced Sentinel-2 image as training data, then employed a RF algorithm with 30 decision trees to classify the image into lake and non-lake. Overall, the RF method has demonstrated significant effectiveness in extracting SGLs, reducing some of the interference from surface runoff and meltwater. To quantitatively evaluate the performance of the classification algorithm, the Intersection over Union (IoU) metric is used, which is the proportion of the overlap between the two results relative to their combined area. Specifically, we manually selected five SGLs on each image, compared them with the RF extraction results, as shown in Fig. 4, and calculated the IoU value for each image. The results are shown in Table 1. The difference between the manually selected continuous boundaries and the jagged edges on raster images significantly affects the accuracy of image IoU. This effect is particularly noticeable during the early melting stages (i.e., June 7) of SGLs since the area of SGLs is relatively small. As the melting intensifies and the area of SGLs increases, the impact of this difference on IoU evaluation decreases in subsequent results, with all IoU values remaining around 90%. Overall, the average IoU value for SGLs across seven periods is 90.20%, providing reliable SGL extents for subsequent experiments.



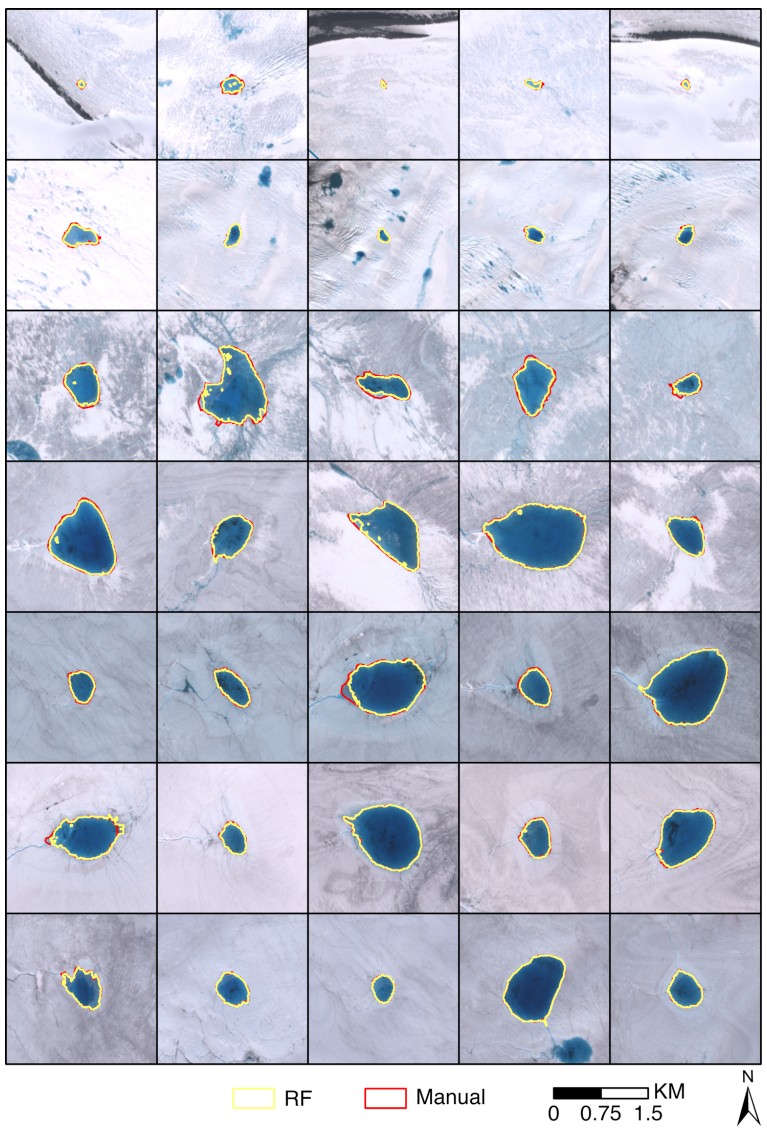

**Figure 4. The first to the seventh rows show the comparison between the extracted extent and manually delineated contours of SGLs from seven different periods between June 7 and August 28, with the background image being the corresponding Sentinel-2 image from each respective period.**

| Date | June 7 | June 17 | July 2 | July 14 | Aug. 1 | Aug. 13 | Aug. 28 | Overall |
|---|---|---|---|---|---|---|---|---|
| **IoU (%)** | 75.55 | 89.44 | 87.39 | 94.77 | 93.30 | 94.74 | 96.24 | 90.20 |

**Table 1. Accuracy assessment of extraction results of SGLs.**

The detection results of the lake surface and bottom are shown in Fig.5. The difference between the two represents the lake's depth. The detected lake depth based on ICESat-2 data is considered the reference in this paper since the reliability of this





method has been verified (Lutz et al., 2024; Melling et al., 2024). In the study area, there are a total of 1991 pixels over 28 SGLs in Sentinel-2 imagery, coinciding with three laser beams (gt1l, gt2l, gt3l) of ICESat-2 RGT No. 338. Among them, 994 pixels are utilized as training samples for the MLP training, while the remaining are used for evaluating the accuracy of MLP

inversion results for lake depth.

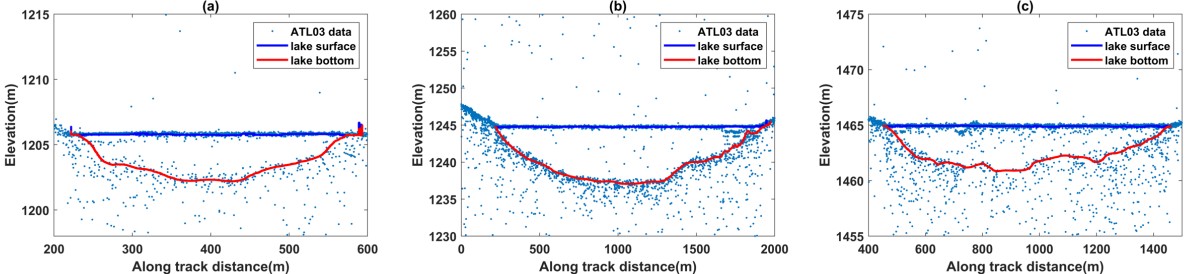

**Figure 5. The lake surface and bottom detection results based on ICESat-2 ATL03 data.**

Additionally, the effectiveness of the empirical formula method (Box and Ski, 2007) in predicting depth, which established an empirical relationship between SGL reflectance and depth as shown in equation (3), is evaluated using the same training and

testing data, allowing for a comparison of the performance between MLP and the empirical formula method.

$$D = \frac{\alpha_0}{R + \alpha_1} + \alpha_2 \tag{3}$$

Where $D$ represents the estimated depth of the SGL. $R$ denotes the reflectance in the green or red band, and parameters $\alpha_0$, $\alpha_1$, and $\alpha_2$ are empirical coefficients fitted by using training data.

The mean absolute error (MAE) is adopted to assess the depth inversion accuracy of both the MLP model and the empirical

formula method. The calculation method for MAE $r_{mean}$ is as equation (4).

$$r_{mean} = \frac{\sum |d_{ref} - d_{pred}|}{N} \tag{4}$$

where $d_{ref}$ represents the lake depth obtained by ICESat-2, $d_{pred}$ represents the predicted lake depth value using the MLP model or empirical formula method, and $N$ is the number of pixels.

The comparison of the depth inversion accuracy for both the empirical formula method and the MLP model over different

depths is presented in Table 2, where the results in bold indicate the highest accuracy in the depth inversion. Although the results based on the empirical formula method demonstrate superiority in predicting depths within the ranges of 1-2 m and 4-6 m, the MLP model exhibits an overall MAE of 0.42 m across all depth ranges, significantly outperforming the empirical formula method. The advantage of the MLP model lies in its ability to leverage multiple inputs for feature selection, which integrates more band information and can fully utilize the information of different bands in the image compared to the single-

band inversion of the empirical formula method, thus achieving higher accuracy results.





| Depth range (m) | Empirical formula method based on Green band (m) | Empirical formula method based on Red band (m) | MLP model (m) |
|---|---|---|---|
| 0-1 | 0.52 | 0.52 | **0.37** |
| 1-2 | 0.52 | **0.30** | 0.43 |
| 2-3 | 0.73 | 0.74 | **0.43** |
| 3-4 | 0.59 | 0.76 | **0.57** |
| 4-5 | **0.18** | 0.22 | 0.19 |
| 5-6 | **0.19** | 0.32 | 0.56 |
| Overall | 0.56 | 0.51 | **0.42** |

**Table 2. MAE of lake depth inversed by the empirical formula method and the MLP model at different depths, where the results in bold indicate the highest accuracy.**

To analyze the distribution of depth inversion errors, we plotted the depth inversion bias maps and assigned different colors
based on the point density (Fig. 6). The depth inversion bias distribution of the MLP method is closer to the horizontal axis
compared to that of the empirical formula method, while the depth inversion results obtained by the empirical formula method
almost always have a certain linear bias. Within the 1-2 m range, precisely where this linear bias predicted by the red band
intersects with the horizontal axis, to some extent explains why the empirical formula method achieves the highest accuracy
in the depth interval of 1-2 m for the red band.

Unlike a physically based depth inversion method, the MLP model learns the relationship between lake depth and input features
through training data. Therefore, issues such as the data quality of the dataset and the uneven distribution of the number of
data samples at each depth will greatly affect the inversion results of the MLP. As can be seen from the depth inversion bias
plot, the number of sample points at depths above 3 m is significantly less than the number of sample points at depths below
3 m, with the lowest number of points at the 5-6 m depth interval. The combination of the above reasons leads to the inversion
results of the MLP model at 5-6 m being inferior to the inversion results of the empirical formula method.



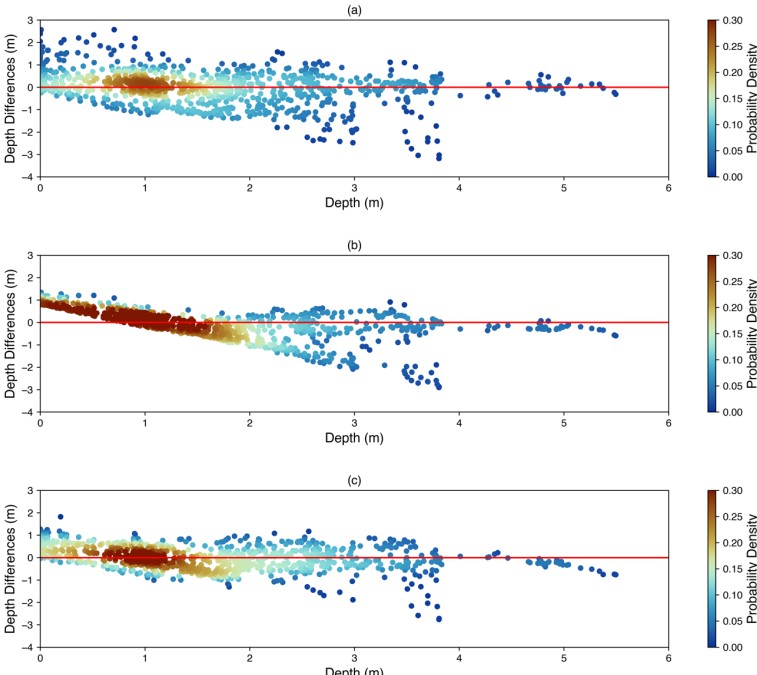

**Figure 6. The depth inversion bias maps obtained by the empirical formula method based on the Green (a) or Red (b) band, and the MLP model (c).**

## 4.2 The spatiotemporal variation characteristics of SGLs' parameters during the 2022 Melt Season

The results of SGLs' area extraction and depth inversion over seven time periods are depicted in Fig. 7. The spatial and temporal distribution of SGLs extracted at different periods shows significant variation. During the entire melt season, SGLs appear at different elevation ranges at different stages, with varying areas and depths. On June 7 and June 17, the SGLs are primarily distributed in the coastal areas around and below an elevation of 800 m (Fig. 7(a)). However, the SGLs on June 7 are smaller in area and shallower in depth, with no significant large lakes. By June 17, the area and depth of the SGLs have increased, and notable lakes appear near the 800 m contour line (Fig. 7(b), Fig. 8(a)). In contrast, the SGLs on July 2 show significant differences from the previous two periods, primarily occupying the 800 m to 1200 m region, with noticeable increases in area and depth (Fig. 7(c), Figs. 8(a)(b)). SGLs larger than $3\times10^6$ m$^2$ begin to appear (red outlines in Fig. 7(c)). On July 14, the SGLs further advanced to higher elevations, extensively distributed between 800 m and 1600 m (Fig. 7(d)). The area and depth of the SGLs continue to grow, with several large lakes around the 1200 m contour line (Fig. 8(b)). The number of lakes larger than $3\times10^6$ m$^2$ increases from one to three (red outlines in Fig. 7(d)). On August 1, the trend of supraglacial advancing to higher elevations slows, with most SGLs distributed between 1200 m and 1600 m (Fig. 7(e)). The overall area and volume slightly increase, with the number of lakes larger than $3\times10^6$ m$^2$ reaching nine (red outlines in Fig. 7(e)). By August 13, the SGLs do not continue to expand to higher elevations, mainly occupying the 1200 m to 1600 m range (Fig. 7(f)). As the



melt season approaches its end, some SGLs are covered by snow and ice, reducing their numbers. However, large and deep SGLs are still around the 1600 m contour line (Fig. 8(c)). On August 28, the number of SGLs significantly decreases, showing some variation in spatial distribution (Fig. 7(g)). A few deep and large lakes remain above 1200 m, while more shallow and small SGLs appear in coastal areas below 800 m.

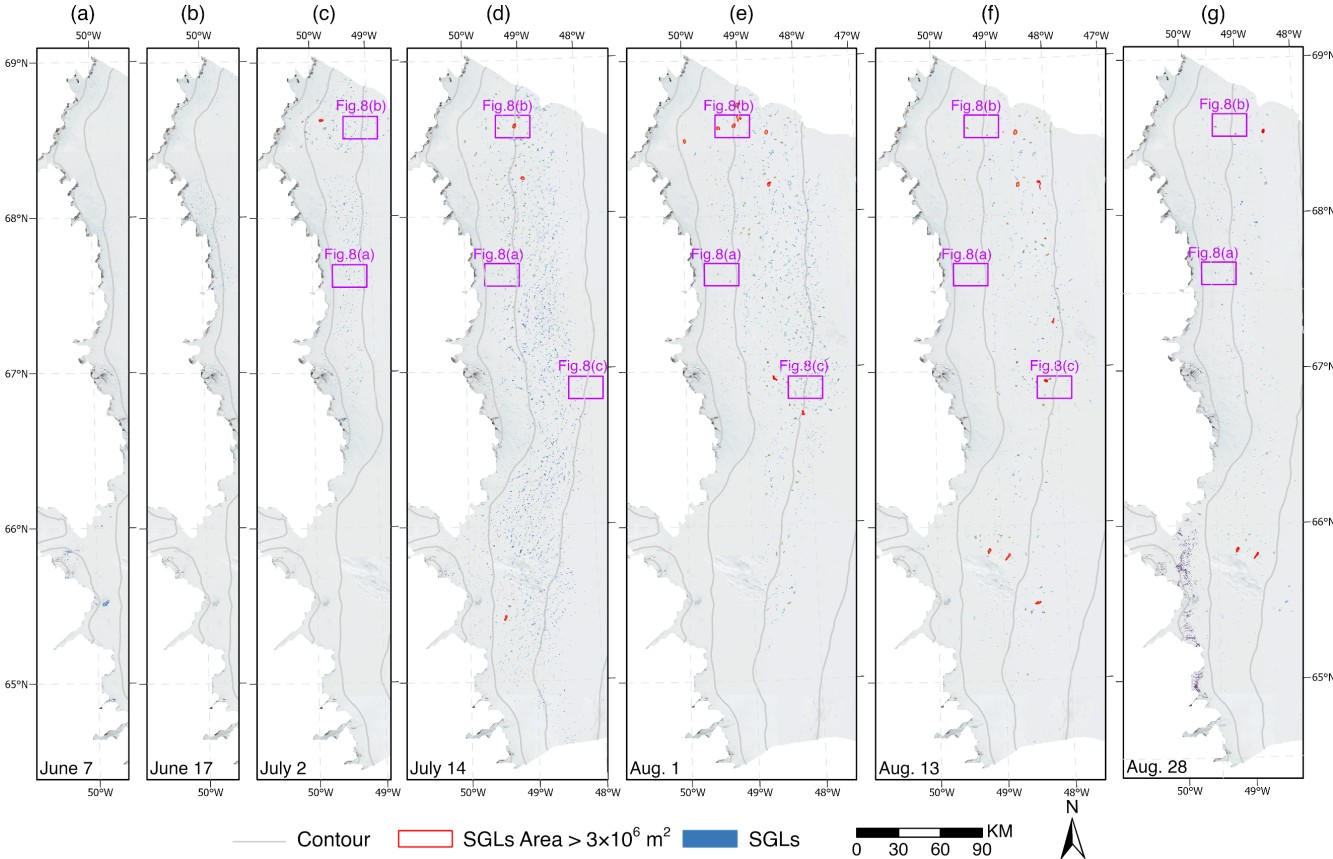

**Figure 7. SGLs' area extraction and depth inversion results on June 7 (a), June 17 (b), July 2 (c), July 14 (d), August 1 (e), August 13 (f), and August 28 (g). The background map is Sentinel-2 images from each respective period.**







**Figure 8. Zoomed-in view on the evolution of SGLs' area and depth at elevations of around 800 m (a), 1200 m (b), and 1600 m (c). The background map is Sentinel-2 images from each respective period.**




To mitigate the impact of differences in the number of available images across different periods, the maximum and average values of area, depth, and volume of SGLs are compared, as shown in Table 3. At the beginning of the development of the SGL (June 7 – June 17), due to the relatively small amount of melting, there are a large number of small pieces of water on
the edge of the ice sheet. At this time, the area, depth, and volume of the individual SGL are relatively small, less than $1\times10^6$ $m^2$ and $1\times10^6$ $m^3$, respectively. Afterward, SGLs enter a period of rapid melting, while the maximum area of the individual lake increases from $0.38\times10^6$ $m^2$ to $3.22\times10^6$ $m^2$ with a growth rate of 748% from June 17 to July 2, and the maximum volume of the SGL increases from $0.52\times10^6$ $m^3$ to $10.60\times10^6$ $m^3$ with a growth rate of 1921%. Similarly, the average area, depth, and volume increased rapidly with a growth rate of 340%, 84%, and 1215%, respectively, which was significantly different from
the previous period and implied that the SGL had begun to enter the peak of its development as the melt season progressed.

After entering the peak period of development, the maximum and mean area of the SGLs show a sustained upward trend between July 2 and July 14, with a growth rate of 49% and 85%, respectively. However, the maximum and mean depth show a decreasing trend. Overall, the maximum and mean volume of the SGLs show an increasing trend, with rises of 28% and 44%, respectively. The growth rate of the area of the SGLs is higher than the growth rate of the volume, indicating a large number
of large and shallower-depth SGLs appeared during this period. Then, the mean area and volume of the SGLs reached the maximum on August 1, with the growth rate of 3% and 68%, respectively, indicating that the mean area of the SGLs has already stabilized at that time, while the mean volume is still in the stage of high-speed growth. On August 13, although the depth of SGLs continues to increase, the area shows a significant decreasing trend, resulting in the volume remaining comparable to that on August 1. Between August 13 and August 28, the maximum and mean values of areas, depth, and volume of SGLs show a decreasing trend, indicating that the lakes begin to recede as time progresses toward the end of the melt season.
And the rate of decline of the mean values of the SGLs (39%) exceeded that of maximum values (23%), indicating that most of the SGLs were frozen or drained overall, but there are a few lakes with large areas and volumes of water that still existed.

| Date | Maximum Area (×10⁶ m²) | Mean Area (×10⁴ m²) | Maximum Depth (m) | Mean Depth (m) | Maximum Volume (×10⁶ m³) | Mean Volume (×10⁴ m³) |
|---|---|---|---|---|---|---|
| June 7 | 0.62 | 0.23 | 4.82 | 0.72 | 0.30 | 0.16 |
| June 17 | 0.38 (-39%) | 0.25 (+9%) | 5.09 (+6%) | 0.88 (+22%) | 0.52 (+75%) | 0.13 (-19%) |
| July 2 | 3.22 (+748%) | 1.10 (+340%) | 5.80 (+14%) | 1.62 (+84%) | 10.60 (+1921%) | 1.71 (+1215%) |
| July 14 | 4.80 (+49%) | 2.04 (+85%) | 5.71 (-2%) | 1.21 (-25%) | 13.54 (+28%) | 2.46 (+44%) |
| Aug. 1 | 6.42 (+34%) | 2.11 (+3%) | 5.71 (0%) | 1.96 (+62%) | 20.93 (+55%) | 4.14 (+68%) |
| Aug. 13 | 5.90 | 1.65 | 5.85 | 2.46 | 21.75 | 4.06 |





| | | | | | |
|---|---|---|---|---|---|
| | (-8%) | (-22%) | (+2%) | (+26%) | (+4%) | (-2%) |
| Aug. 28 | 4.53 | 1.01 | 5.04 | 1.61 | 15.02 | 1.63 |
| | (-23%) | (-39%) | (-14%) | (-35%) | (-31%) | (-60%) |

**Table 3. Statistics of the maximum and mean values of SGLs area, depth, and volume for seven periods, with the changing rate in parentheses.**


The depth distribution of SGLs in each period is plotted as a violin plot (Fig. 9), ignoring lakes above 4 m, which accounted for less than 2% of the total. To demonstrate the difference in distribution between the two adjacent periods, we differentiated the violin plots of the previous time (blue) and the later time (brown) by color, i.e., the distribution of the statistical plots on

the remaining time except the first and last time is compared with its previous and subsequent periods, respectively. The major difference is between the June 17 and July 2 plots, where the peak position is clearly shifted upward and the distribution of depth data between the upper and lower quartiles is more concentrated, i.e., the depth of the SGL begins to develop more steadily on the existing base rather than melting randomly, corresponding to the transition from the early to the peak period of SGLs development. Subsequently, the distribution of maximum depths during the peak period of SGL development is

relatively concentrated with obvious peaks, while the median is similar to the trend of average depths, also appearing to decrease and then increase, indicating that more shallow SGLs are formed during July 2 to 14. The median peak on August 1 and shows the most concentrated distribution shape of all periods, suggesting that the development of SGLs has reached a relative peak and that the maximum depth has stabilized. The subsequent distribution on August 13 differs significantly from that of August 1. Although the median value still increases, there is no longer a prominent peak, and the upper and lower

quartiles show obvious dispersion. The increase in the median value indicates that some deeper SGLs still exist, but the overall dispersion of the data still indicates that the development of SGLs has entered a period of decline, and the distribution of maximum depths on August 28 is more toward shallower water.

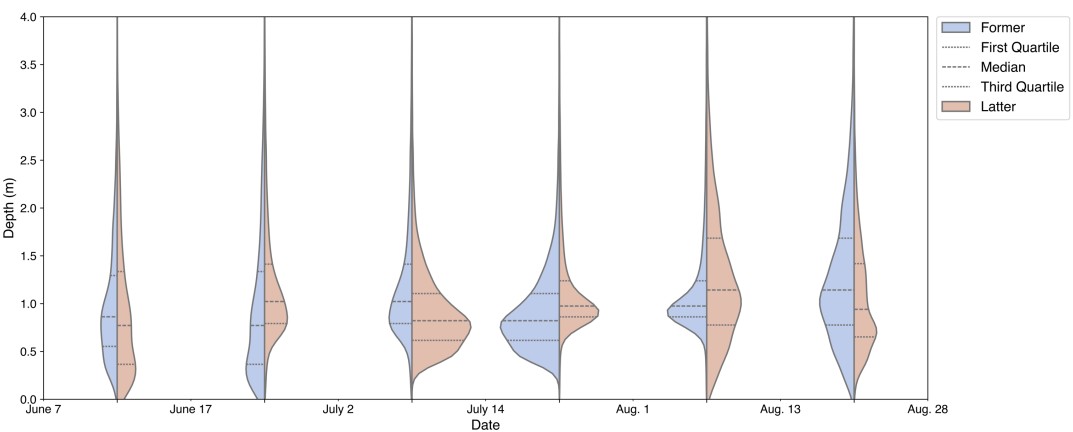

**Figure 9. Violin plots of SGLs' depth distribution at different periods.**





A density map of the distribution of the area of each SGL and its corresponding average depth over different time periods is
created to further examine the distribution and changes in area and depth of SGLs over the melting seasons. For better display
of figures, only the SGLs with an area less than $5\times10^4$ m$^2$ and an average depth within 3 m, which covers more than 90% of
the SGLs within each period, are shown in Fig. 10. Each data point on the density map represents a single SGL, and brighter
colors indicate higher density, corresponding to a greater number of SGLs.

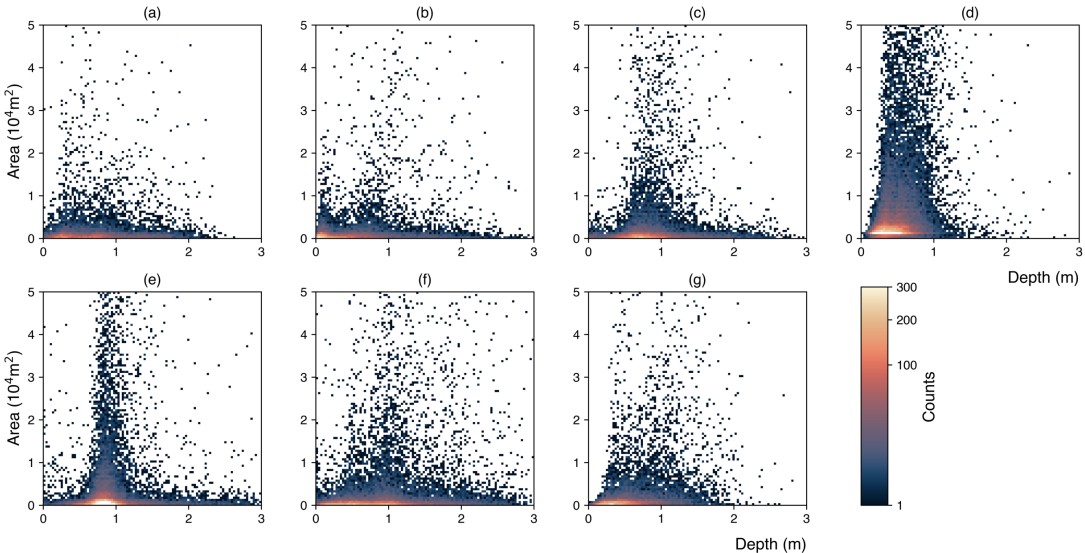


**Figure 10. The Area-Depth distribution map of individual SGL on June 7 (a), June 17 (b), July 2 (c), July 14 (d), August 1 (e),
August 13 (f), and August 28 (g).**

It can be found that the distribution of SGLs across all periods exhibits a clear trend from scattered to concentrated and then
back to scattered. During the initial development phase (as seen in Figs. 10(a)(b)), the brighter regions cluster near the origin,
indicating a higher abundance of small and shallow SGLs. Beginning with July 2, a distinct peak appears, with SGL depths
gradually concentrating between 0.5 m and 1 m, as shown in Fig. 10(c). By July 14, the number of SGLs increases significantly.
A significant portion of these lakes has an average depth distribution in the range of 0.2 m to 1 m, with a pronounced peak
near 0.3 meters, as indicated in Fig. 10(d). In addition, there is a notable increase in the number of SGLs with areas greater
than $2\times10^4$ m² and relatively shallow depths. This observation is consistent with the overall decrease in mean depth during the
development of SGLs as shown in Table 3. On August 1, the distribution of SGLs becomes more concentrated. The brighter
regions shift to deeper mean depths, reaching about 0.8 m, as presented in Fig. 10(e). Many SGLs now have mean depths of
around 0.9 m. Compared to July 14, there is a significant increase in the number of SGLs with mean depths exceeding 2 m.
By August 13, the distinct peak is diminishing, and the distribution of SGLs is no longer concentrated around a single point.
Instead, it gradually spreads out, as shown in Fig. 10(f). The number of lakes with average depths greater than 2 m continues





to increase. At this stage, SGLs show variability, i.e., some evolve into larger, deeper lakes, while others retreat into smaller, shallower lakes. This characteristic marks the transition to the late stage of development. Even within small regions, individual SGLs show significant variation. Within the same area, some lakes may grow larger, while others may freeze or drain (as shown in the fifth and sixth rows in Fig. 7(c)). During the late stage of SGL development, extensive drainage or freezing leads

to reductions in area, depth, and the total number of lakes, as shown in Fig. 10(g). The brighter regions converge to smaller areas and shallower depths, and the number of SGLs with average depths greater than 2 m rapidly decreases.

Finally, Fig. 11 illustrates the volume distributions and the total volume of SGLs at seven periods. The boxplots encapsulate the interquartile range (IQR), with the median volumes denoted by red lines within the turquoise boxes, and the whiskers extending to 1.5 times the IQR. Outliers, which are defined as the first and the last 1% of the data, are indicated by grey points,

while the red triangles connected by a dotted line represent the mean volumes. Throughout the observation period, the median volume of the SGLs exhibits minor fluctuations, the largest median volume is on July 14, which is around 10 m³. Conversely, the mean volumes show a discernible increasing trend from early June, peaking on August 1, before slightly declining towards the end of August. This divergence between the median and mean suggests that while the majority of lakes maintain stable volumes, a subset of lakes experiences significant volume increases, thus elevating the mean. The persistent presence of high-

volume outliers across all dates further corroborates this observation, indicating the existence of lakes with substantially larger volumes compared to the majority. The consistency in the volume range and outliers underscores the dynamic nature of supraglacial lakes, likely influenced by varying melting rates, precipitation, and drainage patterns. This analysis highlights the complex behavior of supraglacial lake volumes over the summer months, with a few lakes significantly impacting the overall mean despite the general stability in median volumes. Meanwhile, the total volume of the SGLs during the seven periods is

also presented by the blue line in Fig. 11. Starting from July 2, the total volume of the SGLs shows a sharp increase, reaching its maximum value of $9.30 \times 10^8$ m³ on August 1. Afterward, the volume of the SGLs begins to decrease until the end of August.

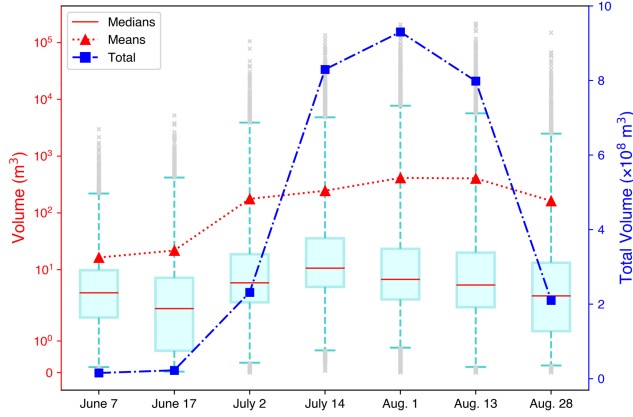

**Figure 11: Boxplots of individual SGLs' volume and the total volume at different time.**



### 4.3 The evolution characteristics of SGLs at Different elevations

To further analyze the spatial distribution of SGL development in relation to elevation, we divided the elevation range from 0 to 2000 m into five intervals of 400 m each and calculated the average area, depth, and volume of SGLs within each interval. Fig. 12 illustrates these statistics, with colors representing elevation from light to dark, gray dashed lines indicating the overall average, and red stars marking the maximum values for each line.

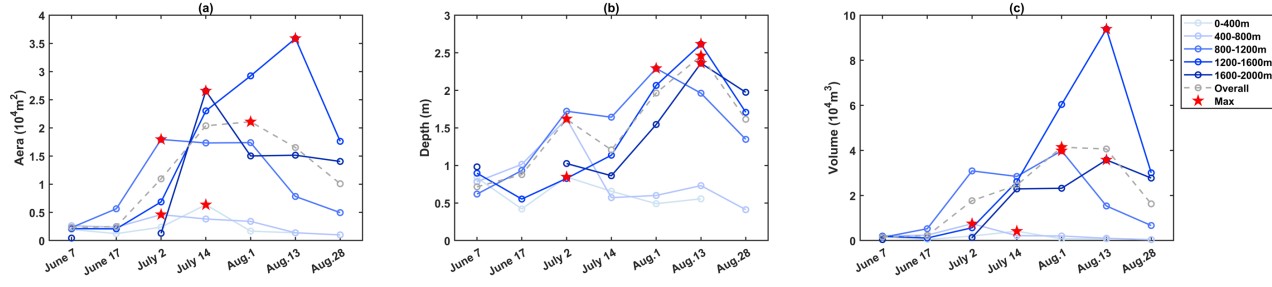

**Figure 12. SGL average area (a), depth (b), and volume (c) at different elevation intervals.**

Generally, there is a significant difference between SGLs above and below 800 m. The average area, depth, and volume of SGLs below 800 m remain relatively stable with minimal fluctuations. In contrast, SGLs above 800 m exhibit more variability and dominate the overall trends of various statistics. The most drastic changes in area, depth, and volume occur between 1200

m and 1600 m, indicating this elevation range is the most favorable for SGL formation.

According to the average area at different elevations (shown in Fig. 12(a)), the overall peak area is reached around early August, specifically on August 1st, driven primarily by lakes in the 1200-1600 m range, with the highest value recorded at $3.5 \times 10^4 \, m^2$ on August 13. Each elevation band shows a similar trend with varying magnitudes, peaking mostly between late July and early August, suggesting mid-summer as the period of maximum lake expansion. After July 14, the area of SGLs between 1200 m

and 1600 m continues to increase until peaking on August 13 and then decreases, while the average area of SGLs above 1600 m decreases and then stabilizes.

Fig. 12(b) presents the average depth of SGLs across different elevations. Peaks in mean depths for the 0-400 m and 400-800 m ranges occurred on July 2, with mean depths in the 400-800 m range significantly higher than those in the 0-400 m range. Before July 2, the overall mean depth change is dominated by SGLs between 400 m and 1200 m. After July 2, the mean depth

change is dominated by SGLs above 800 m. Mean depths in all elevation segments decreased between July 2 and July 14, except for those between 1200 m and 1600 m. Between July 14 and August 1, the mean depths of SGLs above 800 m increased rapidly, creating a more pronounced difference with those below 800 m, with mean depths in the 800-1200 m range reaching a peak. From August 1 to August 13, the mean depth of SGLs above 1200 m continued to increase, albeit at a slower rate, reaching its maximum mean depth on August 13, while the mean depth of SGLs between 800 m and 1200 m began to decrease.

After August 13, the mean depth of SGLs decreased across all elevation bands as the ablation season approached its end.




The average volume of SGLs in different elevation zones generally follows a trend of increasing and then decreasing, with the exception of SGLs between 800 m and 1200 m, which showed a smaller decrease on July 2 and then increased again (as shown in Fig. 12(c)). The higher the elevation of the SGL, the later its average volume reaches its peak, reflecting the spatial distribution of SGLs. As the melt season advances, SGLs gradually push inland from the coast, reaching elevations of 1800 m

or higher. The volume change of SGLs between 1200 m and 1600 m is particularly notable, with the peak average volume significantly larger than that of other elevation intervals. This is due to the cumulative advantage of depth and area, making the probability of large and deep lakes significantly higher in this elevation interval compared to others.

In summary, the elevation range of 1200-1600 m is the most conducive for the development of SGLs, with significant changes in area, depth, and volume, especially during the mid-summer peak. This range sees the most substantial lake formation and

expansion, with the highest occurrence of large and deep lakes.

## 5 Discussion

### 5.1 The uncertainty of the depth inversion of SGL

The presence of ice and snow cover on the surface of SGLs significantly influence the reliability of depth inversion. As shown in Fig. 13(a), a distinct SGL is partially covered with ice or snow. The ICESat-2 track passes over this lake and the covered

areas. In the segment from $P_2$ to $P_3$, the Sentinel-2 image provides band reflection information from the ice surface. This situation presents a significant challenge for both the empirical formula method and the MLP method. As shown in Fig. 13(c), depth measurement methods relying on optical images face obstructions in this segment, leading to an abrupt change in depth and a significant underestimation of depth in ice/snow-covered areas. As for ICESat-2 data, the sparse bathymetric photons, influenced by ice or snow in this segment, as shown in Fig. 13(b), complicates the depth measurements. Although fitting a

continuous bottom (red line in Fig. 13(b)) makes the depth results appear more reasonable, using interpolated depth as a reference does not provide an accurate evaluation of the empirical formula method and the MLP model. Therefore, we manually removed this type of segments to ensure the reliability of depth from ICESat-2 data. In the segment from $P_3$ to $P_4$, although the lake surface is covered by floating ice, the MLP-based depth inversion method can partially overcome the impact of floating ice on the lake surface. This improves the underestimation of lake depth, resulting in depth estimation results closer

to those obtained using ICESat-2 data.

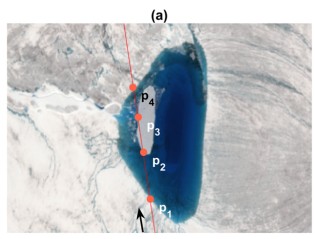
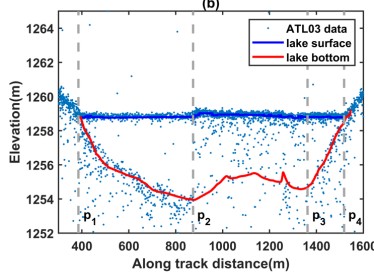
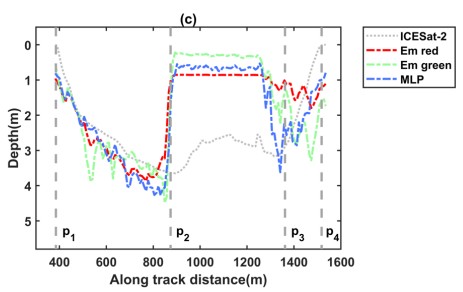





**Figure 13. An example of the uncertainty of depth inversion in ice/snow covered area. (a) Sentinel-2 image obtained on June 17 overlaid with the ICSat-2 RGT 338. (b) The depth detection results by using kernel density estimation. (c) The depth inversion results by using the empirical formula and the MLP model.**

**5.2 The development characteristics of SGLs in the horizontal and vertical directions**

By comparing the change characteristics of the SGLs' average area, depth, and volume over time, we find that there is a difference in the trend of lateral area development and vertical depth development throughout the development of SGLs from their incipient stages to their peak during the melt season (Fig. 14). From the initial melt period of June 7 to July 2, the mean area and the mean depth develop together. However, from July 2 to July 14, the mean area continues to grow, while the mean

depth decreases, implying that more new, shallow SGLs appear. This disparity is the causative factor for the observed trend of a decline in average depth before attaining its peak value. Before reaching the greatest mean depth, SGLs undergo lateral expansion in area, as evidenced by the fact that the rate of increase in mean area was faster than the growth in mean volume from July 2 to July 14. After this period, SGLs develop vertically in depth. From July 14 to August 1, the mean volume continues to grow, reaching its maximum on August 1. This result is similar to the findings of Pitcher and Smith (2019), who

observed that supraglacial streams first incise, resulting in large changes in depth relative to width, and then ablation along channel walls results in lateral expansion, increasing width relative to depth. In addition to the variations in supraglacial lake development resulting from the topography of different regions, further investigation is required to better understand the rate of horizontal and vertical development during the ablation season.

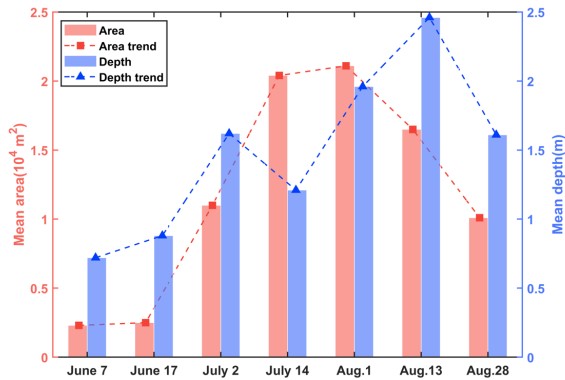

**Figure 14. The trend of lateral area development and vertical depth development throughout the melt season.**

**6 Conclusions**

A method for inversing the volume of SGLs by integrating optical imagery (Sentinel-2) and satellite altimetry data (ICEsat-2) is proposed in this paper. It is indicated that the accuracy of SGLs' area extraction by using RF model based on Sentinel-2 imagery is 90.20%. And the mean absolute error of depth inversion by using a MLP model based on the ratio of reflectance



oriented from Sentinel-2 imagery and the depth of SGLs detected by ICESat-2 data is 0.42 m, surpassing that of traditional empirical formula methods. The proposed volume inversion method for SGLs is applied to southwestern Greenland, thereby obtaining the volumetric evolution of SGLs throughout the entire melt season of 2022. It reveals that SGLs vary significantly in distribution, area, depth, and volume throughout the melt season. The SGLs evolute along coastlines and later spread inland, expanding in area and depth. The maximum of average volume of SGLs is reached on August 1st, amounting to $9.30\times10^8$ m$^3$.

Afterwards, SGLs above 1200 m continue to increase in volume, while SGLs below 1200 m begin to decrease. In late August, as the melt season drew to a close, SGLs diminish and retreat to coastal regions, with a notable reduction in volume. Moreover, the evolution characteristics of SGLs at different elevations are also investigated. It is found that the mean area, mean depth, and mean volume of SGLs below 800 m remains relatively stable throughout the entire melt season. SGLs above 800 m exhibit a similar evolution patten. And the elevation range of 1200 m to 1600 m is the most favorable for the evolution of SGLs.

Moreover, our research indicates a temporal lag in the maximization of mean area and depth, at the onset of development, the area and depth evolve concurrently, then, before the instance when the total volume of meltwater reaches its maximum (Aug. 1), the mean area reaches its peak before the mean depth, suggesting that the SGLs exhibit velocities of morphological evolution along horizontal and vertical dimensions. The quantitative parameter inversion and analysis of SGLs in southwestern Greenland presented in this paper contribute to a better understanding of the mass balance of the Greenland Ice Sheet. However,

when the surface of SGL is covered with ice/snow, the depth may be underestimated, which could further lead to an underestimation of its volume. It may be possible to improve the accuracy of volume estimation by incorporating the temporal changes of the SGL over the time series.

*Code and Data availability*. The code used in this study is available from the corresponding author upon reasonable request.

The NASA Ice, Cloud, and Land Elevation Satellite 2 (ICESat-2) ATL03 v005 data used in this study are publicly available and can be accessed through the National Snow and Ice Data Center (NSIDC) Distributed Active Archive Center (DAAC) at https://nsidc.org/data/icesat-2. The Sentinel-2 data utilized in this research are part of the European Space Agency's (ESA) Copernicus Programme and are freely available for download through the ESA's Sentinel data hub at https://dataspace.copernicus.eu/explore-data/data-collections/sentinel-data/sentinel-2.


*Author contribution*. TF and XM conceptualised the research. TF, XM, and XL designed the study. XM wrote and run of the code, analysed and interpreted of the results. TF and XM prepared the draft of this paper. All the co-authors contributed to paper editing.

*Competing interests*. The authors declare that they have no competing interests.

*Financial support*. This research has been supported in part by the National Key Research & Development Program of China (No. 2021YFB3900105) and in part by the National Science Foundation of China (No. 42371362).



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
