# Peer review of "Volumetric evolution of supraglacial lakes in southwestern Greenland using ICESat-2 and Sentinel-2"

_EGUsphere, 2024_

## Referee Comment (RC1)

Review of "Volumetric evolution of supraglacial lakes in southwestern Greenland using ICESat-2 and Sentinel-2" by Feng et al.

Contents of this review

**General comments**

This paper was an absolute joy to read. I believe it to be a great addition to the literature. In particular, comparing machine learning methods against the more traditional methods will help future researchers and bolster our scientific understanding of lake depth calculation. I have some comments (the vast majority of which are purely technical, so please do not be concerned by the number) that I would like addressed before publication.

**Specific comments (minor)**

**Introduction –**

1.  Line 31-32, you refer to Shepherd et al. (2020) here but there is no corresponding reference in your references list. Please add one.
2.  Line 46, remove "U-net". This is a convolutional neural network and is therefore already included in your list within "convolutional neural networks".
3.  Line 48, I am unsure what you mean here by "favorable outcomes", perhaps give an example or remove this statement as it is currently vague. Suggest removal of "in the area extraction of SGLs" or changing to "SGL area extraction" for readability.
4.  Line 49, change "the area extraction of SGLs" to "SGL area extraction" for readability. Change "observation" to "calculation" as you are not observing.
5.  Line 50, suggest change "reason" to "factor" (personal preference).
6.  Line 54, change "data coverage insufficient" to "data coverage which is insufficient" for readability.
7.  Line 58, Philpot (1987) is the original paper which outlines the radiative transfer equation for calculating supraglacial lake depth. See References section at the end of this review. Remember to remove other Philpot reference from References list once no longer in use.
8.  Line 70-71, suggest rewording from "By measuring the height difference between the surface and bottom photons, the depth of SGLs can be calculated." To "The depth of SGLs can be calculated by

measuring the height difference between the surface and bottom photons." Front-loading this sentence makes it easier to parse.

9.     Line 85, suggest change "utilized" to "used" (personal preference).
10.    Line 93, I think there should be a slight change here unless I am misunderstanding the sentence. I think it should be "The use of machine learning-combined optical images" instead of "The use of machine learning combined optical images" but this is only true if the optical imagery has been combined by machine learning methods. Otherwise, I would suggest altering this sentence to read "The use of machine learning to combine optical images and altimetry data". As it is, this sentence is difficult to understand and needs to be altered in some way.
11.    Line 97, suggest changing "intending to combine" to "leveraging".

**Study area –**

12.    Line 123-124, suggest changing "Sentinel-2 consists of two polar-orbiting satellites (Sentinel-2A and Sentinel-2B), and the dual satellite operation allows Sentinel-2 image data to provide a" to "Sentinel-2 consists of two polar-orbiting satellites (Sentinel-2A and Sentinel-2B), which provides a"
13.    Line 128, change "Considering that SGLs occur only below" to Considering that SGLs only form below"
14.    Line 133-136, I am confused by the two sentences starting, "It should be noted that the" and "Therefore, we use". Are you using multiple images from different days to make a composite image of one day which you then consider to be the 'average' day? You need to rephrase these sentences and make them easier to understand.
15.    Line 139-140, suggest changing from "It can provide elevations of sea ice, land ice, forest canopies, water height, urban areas, etc." to "In achieving this objective, it can provide elevations of sea ice, land ice, and water height amongst other data" – the forest canopies and urban areas are irrelevant for your study.
16.    Line 140-141, change "Equipped with the topographic laser altimetry system Advanced Topographic Laser Altimeter System (ATLAS)" to "Equipped with the advanced topographic laser altimeter system (ATLAS)" – the first part of the sentence is made redundant by the name of the laser altimeter system.
17.    Line 144, please add a citation for this revisit time, I am assuming it is Neumann et al. (2021) but this is not made immediately clear.
18.    Line 146, remove "ellipsoid as the" as it is not necessary.
19.    Line 153, suggest change "utilized" to "used" (personal preference)

**Results –**

20.    Line 232, you mention that there are 28 lakes in this line, but you show (what I presume are) 35 unique lakes in Fig. 4. Is it just that only 28 of these lakes coincide with the ICESat-2 track? If so, you need to change ", coinciding" to "which coincide". At the moment, this is unclear.
21.    Line 234, suggest change "utilized" to "used" (personal preference)
22.    Line 262, reword the sentence beginning "Within the 1-2 m range", it is currently confusing to read and longer than it needs to be.

**Specific comments (major)**

**General to the whole paper –**

23.     I would like to see correct use of past and present tenses throughout the manuscript. As it is, there seems to be some confusion about which one to use in which section.

24.     The manuscript would be made substantially easier to read (and more engaging as a result) by using the active voice. Currently, you have used the passive voice which, although acceptable in scientific writing, is more difficult to read.

**Study area –**

25.     Line 145-155 I have concerns about your use of ATL06. This is derived from ATL03 and doesn't always correctly identify the signal of the ground-return photon events. Please see ATL06 User Guide @ nsidc.org/sites/default/files/documents/user-guide/atl06-v006-userguide.pdf. I would like to see some reference to the limitations of this dataset. My apologies if this is detailed elsewhere in the manuscript and I have simply missed it.

**Methods –**

26.     Line 164, you mention use of the QA60 band to remove cloud and shadow pixels, but the QA60 band is only available on Google Earth Engine for data pre-Feb 2022 and post-Feb 2024. What mask were you using here? I can't see how you could have used this dataset to mask cloud and shadow for your data which is from mid-2022. I am not convinced that cloud and shadow have been appropriately masked considering this.

27.     Equation (2), NDSI is calculated by Hall et al. (1995) with Thematic Mapper (TM) bands 2 and 5 which correspond to Sentinel-2 bands 3 (green) and 11 (SWIR1) not bands 3 and 8 (NIR). Unless you have a good, defendable reason for using NIR in place of SWIR1 here, I suggest recalculating your NDSI and redoing any subsequent analysis. If your training samples are from another time period, you need to specify when they are from (and even if they aren't, you need to be clearer about when they are from).

**Technical corrections**

**Abstract –**

28. Line 9, change "has been the primary" to "is a major"
29. Line 10, change "Greenland Ice Sheet" to "Greenland ice sheet", remove "large amounts of", change "accumulate" to "accumulates"
30. Line 14, change "Random Forest" to "random forest"
31. Line 15, change "Intersection over Union" to "intersection over union"

**Introduction –**

32. Lines 31, 33 & 36, change "Greenland Ice Sheet" to "Greenland ice sheet"
33. Line 42, change "Normalized Difference Water Index" to "normalized difference water index"
34. Line 45-46, change "Random Forest (RF), Support Vector Machine (SVM), U-net, and Convolutional Neural Networks (CNN), have also been employed in the area extraction of SGLs"" to "random forest (RF) algorithm, support vector machines (SVMs), and convolutional neural networks (CNNs), have also been used to extract SGL area" see comment #2 for removal of "U-net".
35. Line 68, change "Ice, Cloud and Land Elevation Satellite-2" to "Ice, Cloud and land Elevation Satellite-2" – the capitalisation is strange on this one, so I have no judgement!
36. Line 70, you're missing a space between "path" and "(Jasinski et al., 2021)"
37. Line 71, change "Lake Surface-Bed Separation" to "lake surface-bed separation"
38. Line 81-82, change "Density-Based Spatial Clustering of Applications with Noise" to "density-based spatial clustering of applications with noise"
39. Line 87, change "Landsat-8" to "Landsat 8", there's no hyphen in this satellite's name
40. Line 96, change "Greenland Ice Sheet" to "Greenland ice sheet"
41. Line 100, change "a MLP" to "an MLP model"
42. Line 105, change both instances of "is" to "are", change "which offering" to "which offer"

**Study area and data –**

43. Line 109 (Title), change from "2 Study Area and Data" to "2 Study area and data"
44. Line 110 (Subtitle), change from "2.1 Study Area" to "2.1 Study area"
45. Line 111, you are missing a space between "Fig." and "1"
46. Line 122, change "2.2 Sentinel-2 Imagery" to "2.2. Sentinel-2 imagery"
47. Line 126, change "Blue" to "blue"
48. Line 127, change "Green" to "green" and "; Red" to "; and red"
49. Line 131-132, change "Top-of-Atmosphere" to "top-of-atmosphere"
50. Line 145, change "ATL03" to "ATLAS version 3 (ATL03)"
51. Line 148, change "ATL06" to "ATLAS version 6 (ATL06)"
52. Line 152, change "Reference Ground Track" to "reference ground track"
53. Line 153, change "it's" to "it is"

**Methods –**

54. Line 159, change "a MLP" to "an MLP"
55. Line 170, change "Normalized Difference Snow Index" to "normalized difference snow index"

56.  Line 171, change "The calculation for *NWDI_{ice}* and *NDSI* is" to "The calculations for *NDWI_{ice}* and *NDSI* are"

57.  Line 176, change "All these" to "All the"

58.  Line 178, change "SGLs' profile" to SGL profile"

59.  Line 187, change "the actual depth of SGL" to either "the actual depth of each SGL" or "the actual depth of the SGL"

60.  Line 191, change "visual inspection. And the" to "visual inspection, and the", and remove "considered as" as this is not necessary

61.  Line 196, change "we construct a" to "we construct an"

62.  Line 200, change "SGL" to "the SGL"

63.  Line 206, change "within seven periods in the whole study area" to "within seven time periods across the whole study area"

**Results –**

64.  Line 214, change "Intersection over Union" to "intersection over union"

65.  Line 219, change "of SGLs" to "of the SGLs" (two instances)

66.  Line 220, change "the area of SGLs" to "the SGL area"

67.  Line 221, change "for SGLs" to "for the SGLs"

68.  Line 230, change "Fig.5. The" to "Fig. 5; the"

69.  Line 246 (Equation 4), the top and bottom spacing of this is different from Equation 3 and may be something you want to change for readability.

70.  Line 259, change "distribution of depth inversion errors, we plotted the depth" to "distribution of the depth inversion errors, we plotted depth"

71.  Line 262, remove "certain"

72.  Line 275, change "SGLs' parameters" to "SGL parameters" and "Melt Season" to "melt season"

73.  Line 276, change "results of SGLs' area" to "results of the SGL area", change "over seven time periods" to "over our studied time periods"

74.  Line 286, I think you may mean "SGLs" instead of "supraglacial" here

75.  Line 303, change "volume of SGLs are compared, as shown in Table 3" to "volume of the SGLs are compared in Table 3"

76.  Line 304, change "SGL" to "SGLs", remove "large", replace "pieces" with "areas"

77.  Line 305, change "SGL" to "SGLs"

78.  Line 306, change "Afterward, SGLs" to "Afterwards, the SGLs", change "while" to "where".

79.  Line 307, I think "748%" should read "+747%" (see comment #156)

80.  Line 308, I think "1921%" should read "+1938%" (see comment #156)

81.  Line 309, change "340%, 84%, and 1215%" to "+340%, +84%, and +1215%"

82.  Line 310, change "implied" to "implies"

83.  Line 312, change "49% and 85%" to "+49% and +85%"

84.  Line 313, change "a decreasing trend" to "decreasing trends", change "28% and 44%" to "+28% and +44%"

85.  Line 314, change "indicating a large number" to "indicating that a large number"

86.  Line 316, change "3% and 68%" to "+3% and +68%"

87.  Line 321, change "(39%)" to "(-39%)", change "(23%)" to "(-23%)"

88.  Line 334, change "SGLs" to "SGL"

89.  Line 337, remove "and"

90.  Line 347, change "within" to "below" (I think this is what you mean? If not, find another way to phrase this as it is currently misleading)

91. Line 354, remove "It can be found that", it is unnecessary.
92. Line 358, change "has" to "have"
93. Line 359, change "meters" to "m"
94. Line 372, change "at seven periods" to "at each of the seven periods"
95. Line 384, change "during the seven periods" to "during the seven study periods"
96. Line 392, change "gray" to "grey", this is how you have spelt it elsewhere in the manuscript

**Discussion –**

97. Line 427 (subtitle), change to "5.1 The uncertainty of SGL depth inversion"
98. Line 438, change "overcome the impact" to "overcome this impact"
99. Line 439, remove "of floating ice on the lake surface", change "underestimation" to "estimation", change "resulting in depth estimation" to "providing"

**Conclusions –**

100. Line 468, change "evolute" to "evolve"
101. Line 469, change "maximum of average" to "maximum total" (I think)
102. Line 474, change "evolution pattern. And" to "evolution pattern, and"
103. Line 475, change "and depth, at the onset" to "and depth. At the onset"
104. Line 477, change "mean depth, suggesting" to "mean depth. This suggests"
105. Line 479, change "Greenland Ice Sheet" to "Greenland ice sheet"
106. Line 480, change "surface of SGL" to "surface of an SGL"

**Figures and Tables**

**Figure 1 –**

107. Please consider making the colour differences more obvious for the number of Sentinel-2 images. They are currently difficult to tell apart.
108. Scale bar, "KM" should be lowercase ("km")
109. Caption, you mention ArcticDEM, is this the mosaic? If so, you need to detail which version and cite the appropriate documentation e.g. "Contour lines calculated from ArcticDEM mosaic version XX are visible as grey lines (CITATION)."

**Figure 2 –**

110. Box "Cloud and shadow removing", change to "Cloud and shadow removal"
111. Box "SGLs Area", change to "SGLs area" (both instances)
112. Box "Region of Interest", change to "Region of interest"
113. Caption, change from "Framework of proposed SGLs' depth inversion method" to "Framework of the proposed SGL depth inversion method"

**Figure 3 –**

114. Labelling, change "Input Features" to "Input features", "Hidden Layer 1" to "Hidden layer 1", "Hidden Layer 2" to "Hidden layer 2", and "Hidden Layer 3" to "Hidden layer 3"
115. Caption, change to "Structure of the MLP model."

**Figure 4 –**

116. My assumption for this figure is that all of the lakes are different, but you need to be explicit about this in the figure caption at the very least.
117. I would suggest labelling the rows with details of each period e.g. Period 1: June 7 etc. This will make it easier to interpret and you won't have to search the paper for details to understand the figure.
118. Scale bar, change "KM" to "km".
119. Caption, remove "The first to seventh rows show" and add "Each row represents a different time period." To the end of the caption.

**Figure 5 –**

120. Are these results indicative, or are they your only results? What do panels (a), (b) and (c) represent? This needs to be explained in your figure caption.
121. Axes labels, each label is missing a space between the label and the unit e.g. "Elevation(m)" instead of "Elevation (m)", please alter these.

**Figure 6 –**

122. Does each point in each of the plots correspond to an SGL pixel? You need to write this in the caption as it is not currently clear where the points are coming from.
123. Caption, "Green" should be "green" and "Red" should be "red".
124. For each panel:

125. Y axis label, change to "Depth difference (m)"
126. Colour bar label, change to "Probability density"
127. I am unfamiliar with probability density but imagine that it has some kind of units, these should be in the colour bar label.

**Figure 7 –**

128. Fig. 8 superimposed boxes, please ensure that these are plotted last as a few are slightly obscured by the SGLs which is making the plot look messy.
129. Fig. 8 superimposed box labels, each of these should be "Fig. 8(X)" not "Fig.8(X)", they are all missing a space after the period.
130. Lat/Lon dashing, either remove this or make it darker. Right now, it is too faded to be of any use to your reader.
131. Legend:
132. Change "SGLs Area" to "SGL area"
133. Change "SGLs" to "SGL" (after blue box)
134. Scale bar, change length to 100 km with markers/intervals at 0, 25, 50, 75 and 100. This will typically be of more use to your reader.
135. Scale bar, change "KM" to "km"
136. Caption, change "SGLs' area" to "SGL area". Where are the contour lines from? I am assuming that they are from ArcticDEM like Fig. 1, but the reader needs to know if it's the mosaic, which version, and have the citation. Please also include the contour distance in the caption too (400 m) e.g. "Contour lines from ArcticDEM mosaic version X are also shown in grey at 400 m intervals (CITATION)."

**Figure 8 –**

137. Labelling (dates), ensure the label spacing is the same for the dates in the corners of your plots as some of the labels are currently going over the bounding boxes (e.g. the 'g' of Aug. 28)
138. Change either the colouring of the depth colour bar to not include red, or change the colour of the SGL area $> 3 \times 10^6$ m$^2$ box as this is currently a bit confusing. I would suggest changing the colour bar to something more colourblind friendly instead of red as you won't have to change previous figures then like you would have to if you changed the red outline colour for SGL area $> 3 \times 10^6$ m$^2$.
139. I would also suggest making the colour bar discrete instead of continuous. You won't lose much definition given that the plots are so small anyway, and it will aid your reader in their interpretation.
140. Change "KM" to "km".
141. Change "SGLs' Area" to "SGL area".
142. Caption, change "on the evolution" to "of the evolution". Where are your contour lines from? Need source, version, citation please.

**Figure 9 –**

*I really liked this way of displaying your data. It's not the most intuitive to interpret from my background, but it explains what you're saying very well and that's what is important.*

143. Key/legend, change "First Quartile" to "First quartile", change "Third Quartile" to "Third quartile"
144. Caption, change to "Figure 9. Violin plots of SGL depth distribution over our seven study periods."

**Figure 10 –**

145. Colour bar label, change to "SGL count"
146. Change "Depth (m)" label placement, currently very strange.
147. Caption, change to "Figure 10. The area-depth distribution map of individual SGLs…"

**Figure 11 –**

148. 2nd Y axis label, change "Total Volume" to "Total volume"
149. Caption, change "SGLs'" to "SGL", change "at different time" to "during the seven study periods"

**Figure 12 –**

150. For each of the August dates, ensure you have a space after the period e.g. "Aug. 1" instead of "Aug.1".

**Figure 13 and Figure 14 –**

151. Y and X axes labels, ensure you have a space after the label and before the unit e.g. "Elevation (m)" instead of "Elevation(m)"

**Table 1 –**

152. Caption, add "showing the intersection over union (IoU) of each time period's SGLs" to the end of the current caption.

**Table 2 –**

153. Column labels, "Green" should be "green" and "Red" should be "red".
154. Caption, add "for each depth range" to the end of the caption.

**Table 3 –**

155. Column labels, change "Maximum Area" to "Maximum area", change "Mean Area" to "Mean area", change "Maximum Depth" to "Maximum depth", change "Mean Depth" to "Mean depth", change "Maximum Volume" to "Maximum volume", change "Mean Volume" to "Mean volume".
156. Percentages, for July 2 maximum area, I get +747%. For June 17 maximum volume, I get +73%. For July 2 maximum volume, I get +1938%. Please check these calculations as I believe they might be wrong. All of the other percentages are correct using this data.
157. Caption, change to "Table 3. Statistics of the maximum and mean values of the SGL area, depth, and volume for the seven study periods, with the growth rate against the previous period given in parentheses."

**References cited within this review**

Hall, D. K., Riggs, G. A., and Salomonson, V. V.: Development of methods for mapping global snow cover using moderate resolution imaging spectroradiometer data, Remote Sensing of Environment, 54, 127–140, https://doi.org/10.1016/0034- 4257(95)00137-P, 1995.

Philpot, W. D.: Radiative transfer in stratified waters: a singlescattering approximation for irradiance, Appl. Optics, 26, 4123–4132, https://doi.org/10.1364/AO.26.004123, 1987

Neumann, T. A., Brenner, A., Hancock, D., Robbins, J., Luthcke, S. B., Harbeck, K., Lee, J., Gibbons, A., Saba, J., and Brunt, K.: ATLAS/ICESat-2 L2A Global Geolocated Photon Data (Version 5), https://doi.org/10.5067/ATLAS/ATL03.005., 2021.

**Final notes**

Although I understand that the number of comments in this review is, quite frankly, atrocious (sorry), the vast majority will not take long to implement and are simply technical. I am contactable for further clarification on any of the comments within this review at l.melling@lancaster.ac.uk.

Please accept my warmest regards and good luck with the changes to this lovely manuscript,

Laura Melling

---

## Referee Comment (RC2)

**Review of "Volumetric evolution of supraglacial lakes in Southwestern Greenland using ICESat-2 and Sentinel-2" by Feng et al.**

This study develops a novel method to estimate the volume of supraglacial lakes during the 2022 melting season in Southwestern Greenland by integrating Sentinel-2 imagery and ICESat-2 data.

**Major comments**

This study employs Sentinel-2 Level-1C images which provide top-of-atmosphere (TOA) reflectance and do not include corrections for atmospheric effects which can lead to biases in the reflectance values and can impact the accuracy of the depth estimation. Was there any atmospheric correction made during the pre-processing of the sentinel-2 imagery?

The paper does not provide enough details about the annotation process for supraglacial lakes. It is unclear how the 50-pixel samples were defined and whether they included a mix of lake and non-lake pixels or if separate negative samples were used. Furthermore, does a sample size of 50 pixels capture the variability in lake sizes? For the evaluation of the random forest model, the method for randomly selecting five lakes is not well explained, and if the five supraglacial lakes chosen per image were excluded from the training dataset. A suggestion would be to split the annotated dataset into training, validation and testing sets instead of randomly selecting supraglacial lakes for the evaluation.

**Minor Comments**

Why is the 2022 melt season selected for this study? Was this melt season used due to significant meltwater?

L 305-306: Area, depth and volume were mentioned, however only area and volume have values, with depth being mentioned with no specific value. Additionally, please clarify "the individual supraglacial lake" further as this is also used in the caption of Figure 10 and figure 11 as well.

**Figrues**

Figure 1: Please include the study period in the caption.
Figure 4: Please include the date range in the figure as well.
Figure 12: The color scheme is difficult to follow. Use more distinguishable color and match the maximum value (star) to the corresponding line color.

---

## Author Comment (AC1)

**egusphere-2024-2195**

Responses to comments from Referee 1

Volumetric evolution of supraglacial lakes in southwestern

Greenland using ICESat-2 and Sentinel-2

We sincerely thank the referee for your constructive comments and suggestions. In the following responses, we use "**bold text**" for the referee's comments, "non-bold" text for our responses, and "*italic*" for text extracted from the manuscript.

**Comments to the Author**

General comments

**This paper was an absolute joy to read. I believe it to be a great addition to the literature. In particular, comparing machine learning methods against the more traditional methods will help future researchers and bolster our scientific understanding of lake depth calculation. I have some comments (the vast majority of which are purely technical, so please do not be concerned by the number) that I would like addressed before publication.**

Response:

> Many thanks for your comments and suggestions. We have carefully considered your comments and made further revisions to enhance the quality of the manuscript. All changes are marked in the revised manuscript, and an item-by-item response to comments is provided in this document.

Specific comments (minor)

**Introduction –**

**1. Line 31-32, you refer to Shepherd et al. (2020) here but there is no corresponding reference in your references list. Please add one.**

Response:

> We have included this reference in the references list of the revised manuscript.

**2. Line 46, remove "U-net". This is a convolutional neural network and is therefore already included in your list within "convolutional neural networks".**

Response:

Removed.

3. **Line 48, I am unsure what you mean here by "favorable outcomes", perhaps give an example or remove this statement as it is currently vague. Suggest removal of "in the area extraction of SGLs" or changing to "SGL area extraction" for readability.**

Response:

We have removed the vague statement and revised the sentence according to the suggestion.

(Line 48 in the marked-up manuscript): "... *These methods demonstrated high accuracy in SGL area extraction. ...*"

4. **Line 49, change "the area extraction of SGLs" to "SGL area extraction" for readability. Change "observation" to "calculation" as you are not observing.**

Response:

Revised.

5. **Line 50, suggest change "reason" to "factor" (personal preference).**

Response:

Revised.

6. **Line 54, change "data coverage insufficient" to "data coverage which is insufficient" for readability.**

Response:

Revised.

7. **Line 58, Philpot (1987) is the original paper which outlines the radiative transfer equation for calculating supraglacial lake depth. See References section at the end of this review. Remember to remove other Philpot reference from References list once no longer in use.**

Response:

We have changed the reference to Philpot (1987), and removed other Philpot reference from References list.

8. **Line 70-71, suggest rewording from "By measuring the height difference between the surface and bottom photons, the depth of SGLs can be calculated." To "The depth of SGLs can be calculated by measuring the height difference between the surface and bottom photons." Front-loading this sentence makes it easier to parse.**

Response:

Revised.

9. **Line 85, suggest change "utilized" to "used" (personal preference).**

Response:

   Revised.

10. **Line 93, I think there should be a slight change here unless I am misunderstanding the sentence. I think it should be "The use of machine learning-combined optical images" instead of "The use of machine learning combined optical images" but this is only true if the optical imagery has been combined by machine learning methods. Otherwise, I would suggest altering this sentence to read "The use of machine learning to combine optical images and altimetry data". As it is, this sentence is difficult to understand and needs to be altered in some way.**

Response:

   What we meant to convey is that the optical images and altimetry data are well combined by the machine learning method. So we have altered the sentence to "The use of machine learning to combine optical images and altimetry data …".

11. **Line 97, suggest changing "intending to combine" to "leveraging".**

Response:

   Revised.

**Study area –**

12. **Line 123-124, suggest changing "Sentinel-2 consists of two polar-orbiting satellites (Sentinel-2A and Sentinel-2B), and the dual satellite operation allows Sentinel-2 image data to provide a" to "Sentinel-2 consists of two polar-orbiting satellites (Sentinel-2A and Sentinel-2B), which provides a".**

Response:

   Revised.

13. **Line 128, change "Considering that SGLs occur only below" to Considering that SGLs only form below".**

Response:

   Revised.

14. **Line 133-136, I am confused by the two sentences starting, "It should be noted that the" and "Therefore, we use". Are you using multiple images from different days to make a composite image of one day which you then consider to be the 'average' day? You need to rephrase these sentences and make them easier to understand.**

Response:

Yes, we use multiple images from different days to achieve complete coverage of the study area, as no single day's imagery could cover the entire study area during the second and third study periods (i.e. 15th–20th June and 30th June– 4th July). As shown in Figure R1, we use multiple images from the same period to achieve complete coverage of the study area. For overlapping regions from different dates, each individual image is processed separately, and the final result is obtained by averaging the processed outcomes.

[Figure]

Figure R1: The coverage area of Sentinel-2 imagery from different dates during (a) the second and (b) third study periods.

We have rephrased these sentences in the revised manuscript.

(Line 138): "... *It should be noted that no single day's imagery could cover the entire study area during the second and third study periods (i.e. June 15–20 and June 30–July 4). Therefore, we use multiple images from different days to achieve complete coverage of the study area, denoted by June 17 and July 2 to represent these two periods in the following text. For overlapping regions from different dates, each individual image is processed separately, and the final result is obtained by averaging the processed outcomes. ...*"

**15. Line 139-140, suggest changing from "It can provide elevations of sea ice, land ice, forest canopies, water height, urban areas, etc." to "In achieving this objective, it can provide elevations of sea ice, land ice, and water height amongst other data" – the forest canopies and urban areas are irrelevant for your study.**

Response:

Revised.

**16. Line 140-141, change "Equipped with the topographic laser altimetry system Advanced Topographic Laser Altimeter System (ATLAS)" to "Equipped with**

**the advanced topographic laser altimeter system (ATLAS)" – the first part of the sentence is made redundant by the name of the laser altimeter system.**

Response:

Revised.

**17. Line 144, please add a citation for this revisit time, I am assuming it is Neumann et al. (2021) but this is not made immediately clear.**

Response:

Yes, the citation Neumann et al. (2021) has been added to the sentence.

**18. Line 146, remove "ellipsoid as the" as it is not necessary.**

Response:

Revised.

**19. Line 153, suggest change "utilized" to "used" (personal preference).**

Response:

Revised.

**Results –**

**20. Line 232, you mention that there are 28 lakes in this line, but you show (what I presume are) 35 unique lakes in Fig. 4. Is it just that only 28 of these lakes coincide with the ICESat-2 track? If so, you need to change ", coinciding" to "which coincide". At the moment, this is unclear.**

Response:

Yes, it means that only 28 lakes coincide with the ICESat-2 track. We have revised the manuscript according to the suggestion.

**21. Line 234, suggest change "utilized" to "used" (personal preference).**

Response:

Revised.

**22. Line 262, reword the sentence beginning "Within the 1-2 m range", it is currently confusing to read and longer than it needs to be.**

Response:

We have rewritten this sentence for clarity.

(Line 279): *".... Within the 1-2 m range, the intersection of the linear bias predicted by the red band with the horizontal axis partly explains why the empirical formula method achieves the highest accuracy in this depth interval. ...."*

**Specific comments (major)**

**General to the whole paper –**

**23. I would like to see correct use of past and present tenses throughout the manuscript. As it is, there seems to be some confusion about which one to use in which section.**

Response:

We have corrected the tenses throughout the manuscript. In the revised version, the past tense is used in the literature review to describe what has been done in previous studies, while the present tense is used in the rest of the sections.

**24. The manuscript would be made substantially easier to read (and more engaging as a result) by using the active voice. Currently, you have used the passive voice which, although acceptable in scientific writing, is more difficult to read.**

Response:

We have reviewed the use of active and passive voice throughout the manuscript and revised some sentences from passive to active voice to improve readability.

**Study area –**

**25. Line 145-155 I have concerns about your use of ATL06. This is derived from ATL03 and doesn't always correctly identify the signal of the ground-return photon events. Please see ATL06 User Guide @ nsidc.org/sites/default/files/documents/user-guide/atl06-v006-userguide.pdf. I would like to see some reference to the limitations of this dataset. My apologies if this is detailed elsewhere in the manuscript and I have simply missed it.**

Response:

Yes, the ATL06 product is derived from ATL03 data. In this study, we first use the ATL06 data, which has a coarse resolution and small data volume, for height noise exclusion as a preprocessing step. Then, we use the ATL03 photon data to perform elevation fitting for both the lake surface and bottom. We have explained this in the revised manuscript.
(Line 156): "… *The ATLAS version 6 (ATL06) surface elevation product at a coarser spatial resolution is used during the preprocessing step to exclude significant height noise.* …
(Line 192): "… *Windows based on ATL06 surface elevation data establish the vertical extent of the ATL03 photon data used, while buffer zones determine the range of data along the track direction.* …

**Methods –**

**26. Line 164, you mention use of the QA60 band to remove cloud and shadow pixels, but the QA60 band is only available on Google Earth Engine for data pre-Feb 2022 and post-Feb 2024. What mask were you using here? I can't see how you could have used this dataset to mask cloud and shadow for your data which is from mid-2022. I am not convinced that cloud and shadow have been appropriately masked considering this.**

Response:

We apologize for the oversight here. We did not remove cloud and shadow pixels due to the absence of the QA60 band. In our study, we use a total of 81 images, 53 of which have 0% cloud cover. Among the remaining 28 images, only one has a cloud coverage rate of 11.09%, while the others exhibit minimal cloud coverage, with an average of 1.62%. This image, acquired on August 28 and belonging to the last study period, is located as shown in Figure R2. According to the Sentinel-2 cloud possibility product downloaded from GEE (https://developers.google.com/earth-engine/datasets/catalog/COPERNICUS_S2_CLOUD_PROBABILITY), areas with a cloud possibility exceeding 60% (indicated in blue area in Figure R2) are primarily located on the right side of the image, at elevations of about 1800–2000 meters. This elevation range corresponds to the critical zone where supraglacial lakes typically form. Based on the development characteristics of supraglacial lakes, during this period, the lakes are retreating from higher to lower elevations as it marks the end of the melt season. Therefore, we conclude that the cloud coverage in this image does not significantly impact the results of our study.

[Figure]

Figure R2: The location of the image with a cloud coverage rate of 11.09%.

We have deleted both the description of cloud and shadow removal and the box of "cloud and shadow removing" in Figure 2 in the revised manuscript.

**27. Equation (2), NDSI is calculated by Hall et al. (1995) with Thematic Mapper (TM) bands 2 and 5 which correspond to Sentinel-2 bands 3 (green) and 11 (SWIR1) not bands 3 and 8 (NIR). Unless you have a good, defendable reason for using NIR in place of SWIR1 here, I suggest recalculating your NDSI and redoing any subsequent analysis. If your training samples are from another time period, you need to specify when they are from (and even if they aren't, you need to be clearer about when they are from).**

Response:

We apologize for our typographical error. We mistakenly wrote "NDWI" as "NDSI". The NDWI, calculated using bands 3 (green) and 8 (NIR), was used as one of the features in the RF model. We have corrected equation (2) and revised the manuscript accordingly.

(Line 178): "… *For feature selection, in addition to the reflection values of the red, green, blue, and NIR bands, NDWIice and NDWI are also included, considering the unique icy and snowy environment of SGLs. The calculations for NDWIice and NDWI are shown in equations (1) and (2), where Rr, Rg, Rb, and RNIR represent the reflection values of the red, green, blue, and near-infrared bands, respectively.*

$$NDWI_{ice} = \frac{R_b - R_r}{R_b + R_r} \qquad\qquad (1)$$

$$NDWI = \frac{R_g - R_{NIR}}{R_g + R_{NIR}} \qquad\qquad (2)$$

…"

**Technical corrections**

**Abstract –**

**28. Line 9, change "has been the primary" to "is a major".**

Response:

Revised.

**29. Line 10, change "Greenland Ice Sheet" to "Greenland ice sheet", remove "large amounts of", change "accumulate" to "accumulates".**

Response:

Revised.

**30. Line 14, change "Random Forest" to "random forest".**

Response:

Revised.

**31. Line 15, change "Intersection over Union" to "intersection over union"**

Response:

Revised.

**Introduction –**

**32. Lines 31, 33 & 36, change "Greenland Ice Sheet" to "Greenland ice sheet".**

Response:

Revised.

**33. Line 42, change "Normalized Difference Water Index" to "normalized difference water index".**

Response:

Revised.

**34. Line 45-46, change "Random Forest (RF), Support Vector Machine (SVM), U-net, and Convolutional Neural Networks (CNN), have also been employed in the area extraction of SGLs" to "random forest (RF) algorithm, support vector machines (SVMs), and convolutional neural networks (CNNs), have also been used to extract SGL area" see comment #2 for removal of "U-net".**

Response:

Revised.

**35. Line 68, change "Ice, Cloud and Land Elevation Satellite-2" to "Ice, Cloud and land Elevation Satellite2" – the capitalisation is strange on this one, so I have no judgement!**

Response:

Revised.

**36. Line 70, you're missing a space between "path" and "(Jasinski et al., 2021)".**

Response:

Revised.

**37. Line 71, change "Lake Surface-Bed Separation" to "lake surface-bed separation".**

Response:

Revised.

**38. Line 81-82, change "Density-Based Spatial Clustering of Applications with Noise" to "density-based spatial clustering of applications with noise".**

Response:

    Revised.

**39. Line 87, change "Landsat-8" to "Landsat 8", there's no hyphen in this satellite's name.**

Response:

    Revised.

**40. Line 96, change "Greenland Ice Sheet" to "Greenland ice sheet".**

Response:

    Revised.

**41. Line 100, change "a MLP" to "an MLP model".**

Response:

    Revised.

**42. Line 105, change both instances of "is" to "are", change "which offering" to "which offer".**

Response:

    Revised.

**Study area and data –**

**43. Line 109 (Title), change from "2 Study Area and Data" to "2 Study area and data".**

Response:

    Revised.

**44. Line 110 (Subtitle), change from "2.1 Study Area" to "2.1 Study area".**

Response:

    Revised.

**45. Line 111, you are missing a space between "Fig." and "1".**

Response:

    Revised.

**46. Line 122, change "2.2 Sentinel-2 Imagery" to "2.2. Sentinel-2 imagery".**

Response:

    Revised.

**47. Line 126, change "Blue" to "blue".**

Response:

Revised.

**48. Line 127, change "Green" to "green" and "; Red" to "; and red"**

Response:

Revised.

**49. Line 131-132, change "Top-of-Atmosphere" to "top-of-atmosphere".**

Response:

Revised.

**50. Line 145, change "ATL03" to "ATLAS version 3 (ATL03)".**

Response:

Revised.

**51. Line 148, change "ATL06" to "ATLAS version 6 (ATL06)".**

Response:

Revised.

**52. Line 152, change "Reference Ground Track" to "reference ground track".**

Response:

Revised.

**53. Line 153, change "it's" to "it is".**

Response:

Revised.

**Methods –**

**54. Line 159, change "a MLP" to "an MLP".**

Response:

Revised.

**55. Line 170, change "Normalized Difference Snow Index" to "normalized difference snow index".**

Response:

Revised.

**56. Line 171, change "The calculation for NWDI$_{ice}$ and NDSI is" to "The calculations for NDWI$_{ice}$ and NDSI are".**

Response:

Revised.

**57. Line 176, change "All these" to "All the".**

Response:

Revised.

**58. Line 178, change "SGLs' profile" to SGL profile".**

Response:

Revised.

**59. Line 187, change "the actual depth of SGL" to either "the actual depth of each SGL" or "the actual depth of the SGL".**

Response:

Revised.

**60. Line 191, change "visual inspection. And the" to "visual inspection, and the", and remove "considered as" as this is not necessary.**

Response:

Revised.

**61. Line 196, change "we construct a" to "we construct an".**

Response:

Revised.

**62. Line 200, change "SGL" to "the SGL".**

Response:

Revised.

**63. Line 206, change "within seven periods in the whole study area" to "within seven time periods across the whole study area".**

Response:

Revised.

**Results –**

**64. Line 214, change "Intersection over Union" to "intersection over union".**

Response:

Revised.

**65. Line 219, change "of SGLs" to "of the SGLs" (two instances).**

Response:

Revised.

**66. Line 220, change "the area of SGLs" to "the SGL area".**

Response:

Revised.

**67. Line 221, change "for SGLs" to "for the SGLs".**

Response:

Revised.

**68. Line 230, change "Fig.5. The" to "Fig. 5; the".**

Response:

Revised.

**69. Line 246 (Equation 4), the top and bottom spacing of this is different from Equation 3 and may be something you want to change for readability.**

Response:

We have adjusted the line spacing so that the top and bottom margins of Equation 4 are the same as those of Equation 3.

**70. Line 259, change "distribution of depth inversion errors, we plotted the depth" to "distribution of the depth inversion errors, we plotted depth".**

Response:

Revised.

**71. Line 262, remove "certain".**

Response:

Revised.

**72. Line 275, change "SGLs' parameters" to "SGL parameters" and "Melt Season" to "melt season".**

Response:

Revised.

**73. Line 276, change "results of SGLs' area" to "results of the SGL area", change**

**"over seven time periods" to "over our studied time periods".**

Response:

Revised.

**74. Line 286, I think you may mean "SGLs" instead of "supraglacial" here.**

Response:

Yes, we have change "supraglacial" to "SGLs".

**75. Line 303, change "volume of SGLs are compared, as shown in Table 3" to "volume of the SGLs are compared in Table 3".**

Response:

Revised.

**76. Line 304, change "SGL" to "SGLs", remove "large", replace "pieces" with "areas".**

Response:

Revised.

**77. Line 305, change "SGL" to "SGLs".**

Response:

Revised.

**78. Line 306, change "Afterward, SGLs" to "Afterwards, the SGLs", change "while" to "where".**

Response:

Revised.

**79. Line 307, I think "748%" should read "+747%" (see comment #156).**

Response:

Yes, it should be +747%. We have double-checked the data calculation, and corrected the mistake. Thanks for your carefulness.

**80. Line 308, I think "1921%" should read "+1938%" (see comment #156).**

Response:

Yes, it should be +1938%. We have double-checked the data calculation, and corrected the mistake. Thanks for your carefulness.

**81. Line 309, change "340%, 84%, and 1215%" to "+340%, +84%, and +1215%".**

Response:

Revised.

**82. Line 310, change "implied" to "implies".**

Response:

Revised.

**83. Line 312, change "49% and 85%" to "+49% and +85%".**

Response:

Revised.

**84. Line 313, change "a decreasing trend" to "decreasing trends", change "28% and 44%" to "+28% and +44%".**

Response:

Revised.

**85. Line 314, change "indicating a large number" to "indicating that a large number".**

Response:

Revised.

**86. Line 316, change "3% and 68%" to "+3% and +68%".**

Response:

Revised.

**87. Line 321, change "(39%)" to "(-39%)", change "(23%)" to "(-23%)".**

Response:

Revised.

**88. Line 334, change "SGLs" to "SGL".**

Response:

Revised.

**89. Line 337, remove "and".**

Response:

Revised.

**90. Line 347, change "within" to "below" (I think this is what you mean? If not, find another way to phrase this as it is currently misleading).**

Response:

Yes, this is exactly what we meant, and we have changed "within" to "below".

**91. Line 354, remove "It can be found that", it is unnecessary.**

Response:

Revised.

**92. Line 358, change "has" to "have".**

Response:

Revised.

**93. Line 359, change "meters" to "m".**

Response:

Revised.

**94. Line 372, change "at seven periods" to "at each of the seven periods".**

Response:

Revised.

**95. Line 384, change "during the seven periods" to "during the seven study periods".**

Response:

Revised.

**96. Line 392, change "gray" to "grey", this is how you have spelt it elsewhere in the manuscript.**

Response:

Revised.

**Discussion –**

**97. Line 427 (subtitle), change to "5.1 The uncertainty of SGL depth inversion".**

Response:

Revised.

**98. Line 438, change "overcome the impact" to "overcome this impact".**

Response:

Revised.

**99. Line 439, remove "of floating ice on the lake surface", change "underestimation" to "estimation", change "resulting in depth estimation" to "providing".**

Response:

Revised.

**Conclusions –**

**100.     Line 468, change "evolute" to "evolve".**

Response:

Revised.

**101.     Line 469, change "maximum of average" to "maximum total" (I think).**

Response:

Yes, we meant the total volume. We have changed "maximum of average" to "maximum total"

**102.     Line 474, change "evolution pattern. And" to "evolution pattern, and".**

Response:

Revised.

**103.     Line 475, change "and depth, at the onset" to "and depth. At the onset".**

Response:

Revised.

**104.     Line 477, change "mean depth, suggesting" to "mean depth. This suggests".**

Response:

Revised.

**105.     Line 479, change "Greenland Ice Sheet" to "Greenland ice sheet".**

Response:

Revised.

**106.     Line 480, change "surface of SGL" to "surface of an SGL".**

Response:

Revised.

**Figures and Tables**

**Figure 1 –**

**107.** **Please consider making the colour differences more obvious for the number of Sentinel-2 images. They are currently difficult to tell apart.**

**108.** **Scale bar, "KM" should be lowercase ("km")**

**109.** **Caption, you mention ArcticDEM, is this the mosaic? If so, you need to detail which version and cite the appropriate documentation e.g. "Contour lines calculated from ArcticDEM mosaic version XX are visible as grey lines (CITATION)."**

Response:

We have made the colour differences more distinct, changed "KM" to "km," and added detailed information about ArcticDEM.

(Line 122):

[Figure]

*Figure 1. Study area. Contour lines calculated from ArcticDEM mosaic version 4.1 (Porter et al., 2023) are visible as grey lines at 400 m intervals. Yellow points indicate the locations of the lakes in the study area, as shown in Fig. 5 and Fig. 13.*

**Figure 2 –**

**110. Box "Cloud and shadow removing", change to "Cloud and shadow removal"**

**111. Box "SGLs Area", change to "SGLs area" (both instances)**

**112. Box "Region of Interest", change to "Region of interest"**

**113. Caption, change from "Framework of proposed SGLs' depth inversion method" to "Framework of the proposed SGL depth inversion method"**

Response:

We have removed box "Cloud and shadow removing" (see response #26 for

reference). The other boxes have revised as suggested.

(Line 170):

[Figure]

*Figure 2. Framework of the proposed SGL depth inversion method.*

**Figure 3 –**

**114. Labelling, change "Input Features" to "Input features", "Hidden Layer 1" to "Hidden layer 1", "Hidden Layer 2" to "Hidden layer 2", and "Hidden Layer 3" to "Hidden layer 3"**

**115. Caption, change to "Structure of the MLP model."**

Response:

We have revised Figure 3 and its caption according to the suggestion.

**Figure 4 –**

**116. My assumption for this figure is that all of the lakes are different, but you need to be explicit about this in the figure caption at the very least.**

**117. I would suggest labelling the rows with details of each period e.g. Period 1: June 7 etc. This will make it easier to interpret and you won't have to search the paper for details to understand the figure.**

**118. Scale bar, change "KM" to "km".**

**119. Caption, remove "The first to seventh rows show" and add "Each row represents a different time period." To the end of the caption.**

Response:

We have revised the caption and revised Figure 4 according to suggestion.

(Line 236):

[Figure]

*Figure 4. The comparison between the extracted extents and manually delineated contours for five different SGLs randomly selected from each study period, using the corresponding Sentinel-2 images as background for each period. Each row represents a different time period.*

**Figure 5 –**

**120. Are these results indicative, or are they your only results? What do panels (a), (b) and (c) represent? This needs to be explained in your figure caption.**

**121. Axes labels, each label is missing a space between the label and the unit e.g. "Elevation(m)" instead of "Elevation (m)", please alter these.**

Response:

These three lakes are indicative, and we have revised the caption. We also added a space between the label and the unit.

(Line 252):

[Figure]

*Figure 5. Three examples of lake surface and bottom detection results based on ICESat-2 ATL03 data. The locations of lakes (a), (b), and (c) are shown in Fig. 1.*

**Figure 6 –**

**122. Does each point in each of the plots correspond to an SGL pixel? You need to write this in the caption as it is not currently clear where the points are coming from.**

**123. Caption, "Green" should be "green" and "Red" should be "red".**

**124. For each panel:**

**125. Y axis label, change to "Depth difference (m)"**

**126. Colour bar label, change to "Probability density"**

**127. I am unfamiliar with probability density but imagine that it has some kind of units, these should be in the colour bar label.**

Response:

Yes, each point in each of the plots corresponds to an SGL pixel. We have added the explanation to the caption, changed the upper case to the lower case, and changed the Y axis label and the colour bar label.

In this study, probability density is calculated using Gaussian kernel density estimation. This method determines the density contribution of each individual sample point to the overall distribution based on its position in the sample space

and its distance from neighbouring sample points. Consequently, the probability density is unitless.

**Figure 7 –**

**128. Fig. 8 superimposed boxes, please ensure that these are plotted last as a few are slightly obscured by the SGLs which is making the plot look messy.**

**129. Fig. 8 superimposed box labels, each of these should be "Fig. 8(X)" not "Fig.8(X)", they are all missing a space after the period.**

**130. Lat/Lon dashing, either remove this or make it darker. Right now, it is too faded to be of any use to your reader.**

**131. Legend:**

**132. Change "SGLs Area" to "SGL area"**

**133. Change "SGLs" to "SGL" (after blue box)**

**134. Scale bar, change length to 100 km with markers/intervals at 0, 25, 50, 75 and 100. This will typically be of more use to your reader.**

**135. Scale bar, change "KM" to "km"**

**136. Caption, change "SGLs' area" to "SGL area". Where are the contour lines from? I am assuming that they are from ArcticDEM like Fig. 1, but the reader needs to know if it's the mosaic, which version, and have the citation. Please also include the contour distance in the caption too (400 m) e.g. "Contour lines from ArcticDEM mosaic version X are also shown in grey at 400 m intervals (CITATION)."**

Response:

We have changed the order of the layers to ensure that the superimposed boxes are at the top layer. We have added a space between "Fig." and "8", changed legend and scale bar, removed the Lat/Lon grid, and revised the caption.

(Line 313):

[Figure]

*Figure 7. SGL area extraction and depth inversion results on June 7 (a), June 17 (b), July 2 (c), July 14 (d), August 1 (e), August 13 (f), and August 28 (g). The background map is Sentinel-2 images from each respective period. Contour lines from ArcticDEM mosaic version 4.1 (Porter et al., 2023) are also shown in grey at 400 m intervals.*

**Figure 8 –**

**137. Labelling (dates), ensure the label spacing is the same for the dates in the corners of your plots as some of the labels are currently going over the bounding boxes (e.g. the 'g' of Aug. 28)**

**138. Change either the colouring of the depth colour bar to not include red, or change the colour of the SGL area > 3 x 10⁶ m² box as this is currently a bit confusing. I would suggest changing the colour bar to something more colourblind friendly instead of red as you won't have to change previous figures then like you would have to if you changed the red outline colour for SGL area > 3 x 10⁶ m².**

**139. I would also suggest making the colour bar discrete instead of continuous. You won't lose much definition given that the plots are so small anyway, and it will aid your reader in their interpretation.**

**140. Change "KM" to "km".**

**141. Change "SGLs' Area" to "SGL area".**

**142. Caption, change "on the evolution" to "of the evolution". Where are your contour lines from? Need source, version, citation please.**

Response:

We have adjusted label positions, modified the depth bar to make it discrete at 1m intervals, changed the legend, and revised the caption.

(Line 320):

[Figure]

*Figure 8. Zoomed-in view of the evolution of SGLs' area and depth at elevations of around 800 m (a), 1200 m (b), and 1600 m (c). The background map is Sentinel-2 images from each respective period. Contour lines from ArcticDEM mosaic version 4.1 (Porter et al., 2023) are also shown in grey at 400 m intervals.*

**Figure 9 –**

**I really liked this way of displaying your data. It's not the most intuitive to interpret from my background, but it explains what you're saying very well and that's what is important.**

**143. Key/legend, change "First Quartile" to "First quartile", change "Third Quartile" to "Third quartile"**

**144. Caption, change to "Figure 9. Violin plots of SGL depth distribution over our seven study periods."**

Response:

Revised.

**Figure 10 –**

**145. Colour bar label, change to "SGL count"**

**146. Change "Depth (m)" label placement, currently very strange.**

**147. Caption, change to "Figure 10. The area-depth distribution map of individual SGLs…"**

Response:

Revised.

**Figure 11 –**

**148. 2nd Y axis label, change "Total Volume" to "Total volume"**

**149. Caption, change "SGLs'" to "SGL", change "at different time" to "during the seven study periods"**

Response:

Revised.

**Figure 12 –**

**150. For each of the August dates, ensure you have a space after the period e.g.**

**"Aug. 1" instead of "Aug.1".**

Response:

    Revised.

**Figure 13 & Figure 14**

**151. Y and X axes labels, ensure you have a space after the label and before the unit e.g. "Elevation (m)" instead of "Elevation(m)"**

Response:

    Revised.

**Table 1**

**152. Caption, add "showing the intersection over union (IoU) of each time period's SGLs" to the end of the current caption.**

Response:

    Revised.

**Table 2**

**153. Column labels, "Green" should be "green" and "Red" should be "red".**

**154. Caption, add "for each depth range" to the end of the caption.**

Response:

    Revised.

**Table 3**

**155. Column labels, change "Maximum Area" to "Maximum area", change "Mean Area" to "Mean area", change "Maximum Depth" to "Maximum depth", change "Mean Depth" to "Mean depth", change "Maximum Volume" to "Maximum volume", change "Mean Volume" to "Mean volume".**

**156. Percentages, for July 2 maximum area, I get +747%. For June 17 maximum volume, I get +73%. For July 2 maximum volume, I get +1938%. Please check these calculations as I believe they might be wrong. All of the other percentages are correct using this data.**

**157. Caption, change to "Table 3. Statistics of the maximum and mean values of**

**the SGL area, depth, and volume for the seven study periods, with the growth rate against the previous period given in parentheses."**

Response:

We have corrected the percentages (see responses #79 and #80 for reference), and revised the caption.

(Line 347):

| Date | Maximum area ($\times 10^6$ m²) | Mean area ($\times 10^4$ m²) | Maximum depth (m) | Mean depth (m) | Maximum volume ($\times 10^6$ m³) | Mean volume ($\times 10^4$ m³) |
|---|---|---|---|---|---|---|
| June 7 | 0.62 | 0.23 | 4.82 | 0.72 | 0.30 | 0.16 |
| June 17 | 0.38 (-39%) | 0.25 (+9%) | 5.09 (+6%) | 0.88 (+22%) | 0.52 (+73%) | 0.13 (-19%) |
| July 2 | 3.22 (+747%) | 1.10 (+340%) | 5.80 (+14%) | 1.62 (+84%) | 10.60 (+1938%) | 1.71 (+1215%) |
| July 14 | 4.80 (+49%) | 2.04 (+85%) | 5.71 (-2%) | 1.21 (-25%) | 13.54 (+28%) | 2.46 (+44%) |
| Aug. 1 | 6.42 (+34%) | 2.11 (+3%) | 5.71 (0%) | 1.96 (+62%) | 20.93 (+55%) | 4.14 (+68%) |
| Aug. 13 | 5.90 (-8%) | 1.65 (-22%) | 5.85 (+2%) | 2.46 (+26%) | 21.75 (+4%) | 4.06 (-2%) |
| Aug. 28 | 4.53 (-23%) | 1.01 (-39%) | 5.04 (-14%) | 1.61 (-35%) | 15.02 (-31%) | 1.63 (-60%) |

*Table 3. Statistics of the maximum and mean values of the SGL area, depth, and volume for the seven study periods, with the growth rate against the previous period given in parentheses.*

---

## Author Comment (AC2)

**egusphere-2024-2195**

Responses to comments from Referee2

Volumetric evolution of supraglacial lakes in southwestern

Greenland using ICESat-2 and Sentinel-2

We sincerely thank the referee for your constructive comments and suggestions. In the following responses, we use "**bold text**" for the referee's comments, "non-bold" text for our responses, and "*italic*" for text extracted from the manuscript.

**This study develops a novel method to estimate the volume of supraglacial lakes during the 2022 melting season in Southwestern Greenland by integrating Sentinel-2 imagery and ICESat-2 data.**

**Major comments**

**This study employs Sentinel-2 Level-1C images which provide top-of-atmosphere (TOA) reflectance and do not include corrections for atmospheric effects which can lead to biases in the reflectance values and can impact the accuracy of the depth estimation. Was there any atmospheric correction made during the pre-processing of the sentinel-2 imagery?**

Response:

In our study, we chose not to perform atmospheric correction to maintain consistency with previous research, including Pope et al. (2016), Williamson et al. (2018), Datta and Wouters (2021), and Melling et al. (2023). Consequently, the potential impact of atmospheric effects on the reflectance values, which could affect depth estimation, was not considered.

Datta, R. T. and Wouters, B.: Supraglacial lake bathymetry automatically derived from ICESat-2 constraining lake depth estimates from multi-source satellite imagery, The Cryosphere, 15, 5115–5132, https://doi.org/10.5194/tc-15-5115-2021, 2021.

Melling, L., Leeson, A., McMillan, M., Maddalena, J., Bowling, J., Glen, E., Sandberg Sørensen, L., Winstrup, M., and Lørup Arildsen, R.: Evaluation of satellite methods for estimating supraglacial lake depth in southwest Greenland, Ice sheets/Glacier Hydrology, https://doi.org/10.5194/tc-2023-103, 2023.

Pope, A., Scambos, T. A., Moussavi, M., Tedesco, M., Willis, M., Shean, D., and Grigsby, S.: Estimating supraglacial lake depth in West Greenland using Landsat 8 and comparison with other multispectral methods, The Cryosphere, 10, 15–27, https://doi.org/10.5194/tc-10-15-2016, 2016.

Williamson, A. G., Banwell, A. F., Willis, I. C., and Arnold, N. S.: Dual-satellite (Sentinel-2 and Landsat 8) remote sensing of supraglacial lakes in Greenland, The Cryosphere, 12, 3045–3065, https://doi.org/10.5194/tc-12-3045-2018, 2018.

**The paper does not provide enough details about the annotation process for supraglacial lakes. It is unclear how the 50-pixel samples were defined and whether they included a mix of lake and non-lake pixels or if separate negative samples were used. Furthermore, does a sample size of 50 pixels capture the variability in lake sizes? For the evaluation of the random forest model, the method for randomly selecting five lakes is not well explained, and if the five supraglacial lakes chosen per image were excluded from the training dataset. A suggestion would be to split the annotated dataset into training, validation and testing sets instead of randomly selecting supraglacial lakes for the evaluation.**

Response:

Thanks for these questions. To extract SGLs from Sentinel-2 images using the RF model, we randomly selected 50 pixels of SGLs and 50 pixels of non-SGL areas from each set of images across different time periods. As a result, the training dataset comprised 350 positive samples and 350 negative samples. To ensure diversity of the positive samples, both typical lake water pixels and those from the edges where ice meets water were included. The number of selected pixels was not influenced by size variations of the SGLs, as the classification process operated on a pixel-based level. For classification evaluation, five lakes from each time period excluded from the training samples were selected to assess accuracy. Additionally, the Intersection over Union (IoU) metric was employed to evaluate the completeness of extraction for entire lake regions. Unlike methods that split samples into training and validation sets, this approach enables an assessment of overall lake region extraction, rather than solely focusing on the classification accuracy of individual pixels.

We have clarified the selection of the training samples.

(Line 223 in the marked-up manuscript): "… *For each time period, we randomly sample 50 pixels from SGL areas and 50 pixels from other areas in the mosaiced Sentinel-2 image as training data, …*"

**Minor comments**

**Why is the 2022 melt season selected for this study? Was this melt season used due to significant meltwater?**

Response:

Studies have shown that surface meltwater in Greenland has been increasing in recent years (Shepherd et al., 2020; Slater et al., 2021). Therefore, we selected the recent year of 2022 to investigate the characteristics of supraglacial lake variations

throughout the entire melt season, as their formation and drainage play a significant role in influencing surface meltwater dynamics.

Shepherd, A., Ivins, E., Rignot, E., Smith, B., van den Broeke, M., Velicogna, I., Whitehouse, P., Briggs, K., Joughin, I., Krinner, G., Nowicki, S., Payne, T., Scambos, T., Schlegel, N., Geruo, A., Agosta, C., Ahlstrom, A., Babonis, G., Barletta, V. R., Bjork, A. A., Blazquez, A., Bonin, J., Colgan, W., Csatho, B., Cullather, R., Engdahl, M. E., Felikson, D., Fettweis, X., Forsberg, R., Hogg, A. E., Gallee, H., Gardner, A., Gilbert, L., Gourmelen, N., Groh, A., Gunter, B., Hanna, E., Harig, C., Helm, V., Horvath, A., Horwath, M., Khan, S., Kjeldsen, K. K., Konrad, H., Langen, P. L., Lecavalier, B., Loomis, B., Luthcke, S., McMillan, M., Melini, D., Mernild, S., Mohajerani, Y., Moore, P., Mottram, R., Mouginot, J., Moyano, G., Muir, A., Nagler, T., Nield, G., Nilsson, J., Noel, B., Otosaka, I., Pattle, M. E., Peltier, W. R., Pie, N., Rietbroek, R., Rott, H., Sorensen, L. S., Sasgen, I., Save, H., Scheuchl, B., Schrama, E., Schroder, L., Ki-Weon Seo, Simonsen, S. B., Slater, T., Spada, G., Sutterley, T., Talpe, M., Tarasov, L., van de Berg, W. J., van der Wal, W., van Wessem, M., Vishwakarma, B. D., Wiese, D., Wilton, D., Wagner, T., Wouters, B., and Wuite, J.: Mass balance of the Greenland Ice Sheet from 1992 to 2018, Nature, 579, 233–9, https://doi.org/10.1038/s41586-019-1855-2, 2020.

Slater, T., Shepherd, A., McMillan, M., Leeson, A., Gilbert, L., Muir, A., Munneke, P. K., Noël, B., Fettweis, X., van den Broeke, M., and Briggs, K.: Increased variability in Greenland Ice Sheet runoff from satellite observations, Nat Commun, 12, 6069, https://doi.org/10.1038/s41467-021-26229-4, 2021.

**L 305-306: Area, depth and volume were mentioned, however only area and volume have values, with depth being mentioned with no specific value. Additionally, please clarify "the individual supraglacial lake" further as this is also used in the caption of Figure 10 and figure 11 as well.**

Response:

We have specified the value of depth. The term 'individual supraglacial lake' refers to each separate supraglacial lake. To ensure clarity, we have revised the words to "each individual SGL" in this sentence as well as in the caption for Figures 10 and 11.

(Line 327): "… *At this time, the area and volume of each individual SGL are relatively small, measuring less than $1 \times 10^6$ m² and $1 \times 10^6$ m³, respectively, with a mean depth of less than 1 m. …*"

**Figures**

**Figure 1: Please include the study period in the caption.**

Response:

We have added the study period in the caption of Figure 1.

(Line 123): "*Figure 1. Study area. Contour lines calculated from ArcticDEM mosaic version 4.1 (Porter et al., 2023) are visible as grey lines at 400 m intervals. Yellow points indicate the locations of the lakes in the study area, as shown in Fig. 5 and Fig. 13. The study period is from June to August 2022.*"

**Figure 4: Please include the date range in the figure as well.**

Response:

We have added the date in the Figure 4.

(Line 237): "

[Figure]

*Figure 4. The comparison between the extracted extents and manually delineated contours for five different SGLs randomly selected from each study period, using the corresponding Sentinel-2 images as background for each period. Each row represents a different time period.* "

**Figure 12: The color scheme is difficult to follow. Use more distinguishable color and match the maximum value (star) to the corresponding line color.**

Response:

We have changed the colors of the lines and the stars.

(Line 424):

---

## Referee Report (RR1)

**Re-review of the manuscript of "Volumetric evolution of supraglacial lakes in southwestern Greenland using ICESat-2 and Sentinel-2"**

I thank the authors for their revisions and addressing the concerns raised in the first round of review. The manuscript has improved significantly; however, I still have some concerns which are detailed below.

Major comment:

**Section 3.1.** Shadows can be misclassified as supraglacial lakes in the optical satellite imagery since they have similar spectral properties. I noticed that the section (3.1.) describing cloud and shadow removal was removed from the current version of the manuscript.
It would be helpful if the authors could clarify how they addressed shadow misclassification, especially considering the limited number of lake pixels used for training, which might not capture the spatial variability of lakes to distinguish between shadows and lakes.

Minor comments:

I suggest moving the sampling strategy for RF classification, the explanation of IoU and MAE evaluation metrics, and formulas 3 and 4 from the Results section to the Methods. It would be clearer to know this information before the results are presented.

I recommend including a cross-validation or comparison with in situ supraglacial lake depth measurements from other studies. For example, the study by Lutz et al. (2024), which is already cited in the manuscript and provides direct lake depth measurements from Northeast Greenland using a remote-controlled sonar boat. The data is also publicly available:
Lutz, Katrina; Bever, Lily; Sommer, Christian; Seehaus, Thorsten; Humbert, Angelika; Scheinert, Mirko; Braun, Matthias Holger (2024): In situ supraglacial lake depth measurements using remote controlled sonar boat in Northeast Greenland, July 2022 [dataset]. PANGAEA, https://doi.org/10.1594/PANGAEA.971782

---

## Author Response (AR2)

**egusphere-2024-2195**

Responses to comments from Referee 2

Volumetric evolution of supraglacial lakes in southwestern

Greenland using ICESat-2 and Sentinel-2

We thank editor for your constructive comments and suggestions. In the following responses, we use "**bold text**" for editor's comments, "non-bold" text for our responses, and "*italic*" for text extracted from the manuscript.

**Major comment:**

**Section 3.1. Shadows can be misclassified as supraglacial lakes in the optical satellite imagery since they have similar spectral properties. I noticed that the section (3.1.) describing cloud and shadow removal was removed from the current version of the manuscript.**

**It would be helpful if the authors could clarify how they addressed shadow misclassification, especially considering the limited number of lake pixels used for training, which might not capture the spatial variability of lakes to distinguish between shadows and lakes.**

Response:

Thanks for the suggestion. We adopted reviewer 1's suggestion to remove the cloud-removal description from the original version of the manuscript. On one hand, the QA60 band is unavailable in Google Earth Engine (GEE) for Sentinel-2 data during our study period (mid-2022). On the other hand, cloud coverage in the images we used was not significant. Therefore, we did not perform cloud and shadow removal in this study.

Specifically, we use a total of 81 images in this study, 53 of which have 0% cloud cover. Among the remaining 28 images, only one has a cloud coverage rate of 11.09%, while the others exhibit minimal cloud coverage, with an average of 1.62%. This image, acquired on August 28 and belonging to the last study period, is located as shown in Figure R1. According to the Sentinel-2 cloud possibility product downloaded from GEE (https://developers.google.com/earth-engine/datasets/catalog/COPERNICUS_S2_CLOUD_PROBABILITY), areas with a cloud possibility exceeding 60% (indicated in blue area in Fig. R1) are primarily located on the right side of the image, at elevations of about 1800–2000 m. Based on the development characteristics of supraglacial lakes, during this period, the lakes are retreating from higher to lower elevations as it marks the end of the melt season. Therefore, we conclude that the cloud coverage in this image

does not significantly impact the results of this study. However, if future studies use images with extensive cloud coverage, careful consideration of shadow misclassification issues will be necessary.

[Figure]

Fig. R1 The location of the image with a cloud coverage rate of 11.09%.

We have added the description regarding cloud coverage in the images used in this study.

(Line 131 in the marked-up manuscript): "… *In this study, we use a total of 81 images in this study, 53 of which have 0% cloud cover. Among the remaining 28 images, only one has a cloud coverage rate of 11.09%, while the others exhibit minimal cloud coverage, with an average of 1.62%. …*"

**Minor comments:**

**I suggest moving the sampling strategy for RF classification, the explanation of IoU and MAE evaluation metrics, and formulas 3 and 4 from the Results section to the Methods. It would be clearer to know this information before the results are presented.**

Response:

Thanks for the suggestion. We have moved the RF sampling strategy for RF classification to the Methods section. We have also added Section 3.4 to Method Section, including the explanation of IoU and MAE evaluation metrics, and the empirical formula approach for depth estimation.

(Line 178): "… *For each time period, we randomly sample 50 pixels from SGL areas and 50 pixels from other areas in the mosaiced Sentinel-2 image as training data, then employ an RF algorithm with 30 decision trees to classify the image into lake and non-lake. …*"

(Line 215): "*3.4 Evaluation methods*

*To quantitatively evaluate the performance of the classification algorithm, the intersection over union (IoU) metric is used, which is the proportion of the overlap between the classification results and manually selected SGLs relative to their combined area. To evaluate the performance on depth prediction, we compare the effectiveness of both MLP model and the empirical formula approach (Box and Ski, 2007). The empirical formula approach establishes an empirical relationship between SGL reflectance and depth as shown in equation (3). The same training and testing data is adopted, allowing for a comparison of the performance between MLP and the empirical formula method.*

$$D = \frac{\alpha_0}{R + \alpha_1} + \alpha_2 \qquad (3)$$

*Where D represents the estimated depth of the SGL. R denotes the reflectance in the green or red band, and parameters $\alpha_0$, $\alpha_1$, and $\alpha_2$ are empirical coefficients fitted by using training data.*

*The mean absolute error (MAE) is adopted to assess the depth inversion accuracy of both the MLP model and the empirical formula method. The calculation method for MAE $r_{mean}$ is as equation (4).*

$$r_{mean} = \frac{\sum |d_{ref} - d_{pred}|}{N} \qquad (4)$$

*Where $d_{ref}$ represents the lake depth obtained by ICESat-2, $d_{pred}$ represents the predicted lake depth value using the MLP model or empirical formula method, and N is the number of pixels. …*"

**I recommend including a cross-validation or comparison with in situ supraglacial lake depth measurements from other studies. For example, the study by Lutz et al. (2024), which is already cited in the manuscript and provides direct lake depth measurements from Northeast Greenland using a remote-controlled sonar boat. The data is also publicly available:**

**Lutz, Katrina; Bever, Lily; Sommer, Christian; Seehaus, Thorsten; Humbert, Angelika; Scheinert, Mirko; Braun, Matthias Holger (2024): In situ supraglacial lake depth measurements using remote controlled sonar boat in Northeast Greenland, July 2022 [dataset]. PANGAEA, https://doi.org/10.1594/PANGAEA.971782**

Response:

Thanks for the suggestion. We have downloaded the relevant in-situ data from the links you provided, including bathymetric data for three SGLs (Lake 522 and Lake 610b measured on July 9[th] 2022, and Lake 610a measured on July 4[th] 2022). The locations of these three SGLs are shown in Fig. R2(a). We have also downloaded the corresponding Sentinel-2 imagery for these dates, as shown in Figs. R2(b) and (c). However, due to heavy cloud cover on July 9[th] (Fig. R2(c)), Lakes 522 and

610b are not visible in the Sentinel-2 imagery. Therefore, we evaluate our results using only the bathymetric data from Lake 610a.

The comparison between the in-situ SGL depth measurements and the depth predictions from our MLP model is shown in Fig. R3. The results indicate good agreement for depth below 3 m (with an MAE of 0.50 m), while deviations become more pronounced for depths exceeding 3 m. We attribute these deviations to the limited training samples available for the MLP model. Considering that the validation data is from a region far from our study area (Fig. R2(a)), we have not included these evaluation results in the manuscript.

[Figure]

Fig. R2 (a) Location of in-situ depth measurements of SGLs (red star) and study area (red polygon). (b) Sentinel-2 imagery obtained on July 4th 2022. (c) Sentinel-2 imagery obtained on July 9th 2022.

[Figure]

Fig. R3 The comparison of in-situ depth measurements and depth prediction results.